



**Climate and basin drivers of seasonal river water**
**temperature dynamics**
**C. L. R. Laizé[1, 2], C. Bruna Meredith[2,*], M. Dunbar[1,**], D.M. Hannah[2]**
[1]{Centre for Ecology and Hydrology, UK}
[2]{School of Geography, Earth, & Environmental Sciences, University of Birmingham, UK}
[*]{now at: Scottish Environment Protection Agency, UK}
[**]{now at: Environmental Agency, UK}
Correspondence to: C. L. R. Laizé (clai@ceh.ac.uk)
**Abstract**
Stream water temperature is a key control of many river processes (e.g. ecology,
biogeochemistry, hydraulics) and services (e.g. power plant cooling, recreational use).
Consequently, the effect of climate change and variability on stream temperature is a major
scientific and practical concern. This paper aimed (1) to improve the understanding of large-
scale spatial and temporal variability in climate–water temperature associations, and (2) to
assess explicitly the influence of basin properties as modifiers of these relationships. A dataset
was assembled including six distinct modelled climatic variables (air temperature, downward
shortwave and longwave radiation, wind speed, specific humidity, and precipitation) and
observed stream temperatures for the period 1984–2007 at 35 sites located on 21 rivers within
16 basins (Great Britain geographical extent); the study focused on broad spatio-temporal
patterns hence was based on three-month averaged data (i.e. seasonal). A wide range of basin
properties was derived. Five models were fitted (all seasons, winter, spring, summer, and
autumn). Both site and national spatial scales were investigated at once by using multi-level
modelling with linear multiple regressions. Model selection used Multi-Model Inference,
which provides more robust models, based on sets of good models, rather than a single best
model. Broad climate-water temperature associations common to all sites were obtained from
the analysis of the fixed coefficients, while site-specific responses, i.e. random coefficients,
were assessed against basin properties with ANOVA. All six climate predictors investigated
play a role as a control of water temperature. Air temperature and shortwave radiation are
important for all models/ seasons, while the other predictors are important for some models/



seasons only. The form and strength of the climate-stream temperature association vary
depending on season and on water temperature. The dominating climate drivers and physical
processes may change across seasons, and across the stream temperature range. The role of
basin permeability, size, and elevation as modifiers of the climate-water temperature
associations was confirmed; permeability has the primary influence, followed by size and
elevation. Smaller, upland, and/or impermeable basins are the most influenced by atmospheric
heat exchanges, while larger, lowland and permeable basins are least influenced. The study
showed the importance of accounting properly for the spatial and temporal variability of
climate-stream temperature associations and their modification by basin properties.
**1    Introduction**
River and stream water temperature (WT) is a key control of many river processes (e.g.
ecology, biogeochemistry, hydraulics) and services (e.g. power plant cooling, recreational
use); Webb et al. (2008). From the perspective of river ecology, WT's influence is both
direct—e.g. organism growth rates (Imholt et al., 2013), predator-prey interactions (Boscarino
et al., 2007), activity of poikilotherms, geographical distribution (Boisneau et al., 2008)—and
indirect, e.g. water quality (chemical kinetics), nutrient consumption, food availability
(Hannah and Garner, 2015).
Consequently, the effect of climate change and variability on stream temperature is a major
scientific and practical concern (Garner et al., 2014). River thermal sensitivity to climate
change and variability is controlled by complex drivers that need to be unravelled (a) to better
understand patterns of spatio-temporal variability and (b) the relative importance of different
controls to inform water and land management, especially climate change mitigation and
adaptations strategies (Hannah and Garner, 2015). There is a growing body of river
temperature research but there is still limited understanding of large-scale spatial and
temporal variability in climate–WT associations, and of the influence of basin properties as
modifiers of these relationships (Garner et al., 2014). This paper capitalises on the PhD
research carried out by the first author (Laizé, 2015).
River thermal regimes are complex because they involve many interacting drivers (Hannah et
al., 2004, 2008). Caissie (2006) identified atmospheric conditions as the primary group of
controls, with hydrology linked to basin physical properties (e.g. topography, geology) as
secondary influencing factors.



The main climate variables (Fig. 1) which constitute an 'atmospheric conditions' group, can
be identified by analysing the theoretical heat budget for a stream reach without tributary
inflow, which may be expressed as (adapted from Hannah and Garner, 2015):
$Q_n = Q* + Q_h + Q_e + Q_{bhf} + Q_f + Q_a$            Equation 1
where $Q_n$ is the total net heat exchange, $Q*$ the heat flux due to net radiation, $Q_h$ the heat flux
due to sensible transfer between air and water (sensible heat), $Q_e$ the heat flux due to
evaporation and condensation (latent heat), $Q_{bhf}$ the heat flux to and from the river bed, $Q_f$ the
heat flux due to friction at the bed and banks, and $Q_a$ the heat flux due to advective transfer by
precipitation and groundwater.
The different components of Eq. (1) correspond to different processes, related to climatic and
hydrological conditions. $Q*$ corresponds to shortwave radiation (insolation from the sun) and
longwave radiation (emitted towards the stream by clouds and overhanging surfaces such as
vegetation, and reemitted back to space (lost) at water surface temperature). Qh corresponds
to convective energy exchanges between air and water (at the surface) causing heat loss or
gain. Qe represents heat loss by evaporation or gain by condensation. Qbhf and Qf do not
relate directly to climate processes but rather local hydrological conditions. (Qf can be
assumed to be negligible in many systems; e.g. Hannah et al., 2008). Qa corresponds to
advective heat exchanges, e.g. inflow or outflow into the river reach, hyporheic exchange,
groundwater. A direct, climatic component of Qa is precipitation inputs, which is thought to
have a limited contribution (Caissie, 2006).
These variables are not independent. Figure 1 features a schematic representation of the
interactions between these variables. Downward short and long wave radiations increase WT
but also air temperature, then there are exchanges between air and water, to influence sensible
heating. Additionally, wind plays a significant role by increasing evaporative cooling and in
modifying the air–water exchanges by increasing mixing (Hannah et al., 2008). The physical
equations underpinning the role of wind can be found in Caissie et al. (2007).
A review of recent international water temperature research can be found in Hannah and
Garner (2015). To date most, UK-focused studies (Table 1) tend to be either specific to a few
monitoring sites, to have a limited geographical extent (i.e. focused with specific region of the
country), and /or to consider few climate drivers. In addition, seasonality is only explored
formally in a small number of papers (e.g. Langan et al., 2001 ). A major difficulty is to pair
WT and climate monitoring sites, as monitoring is coordinated rarely, then to identify time




series with long enough common periods of record. For example, Garner et al. (2014)
undertook a England and Wales scale study and matched water temperature monitoring sites
with climate and hydrological monitoring sites for 38 temperature sites out of ~ 3,000 sites in
the Environment Agency's Freshwater Temperature Archive (Orr et al., 2014). Garner et al.
(2014) is one of the very few studies (internationally) to consider explicitly the role of a
limited number of basin properties.
In most of these studies, analyses are done on a site by site basis, which limits the extent to
which broad patterns can be inferred (statistical results for a given site are only valid for that
site); Caissie, 2006 emphasized this as a limitation when having to work across different
spatial scales. In contrast, studies like Garner et al. (2014) group sites together using
classification techniques to identify regional patterns. However, doing so causes a loss of
information since data-points of all sites within a class are summarised and intra-class
differences lost, and inferences at group level are not necessarily valid at site level. An
alternative analytical/ statistical method, which can characterise broad patterns while
preserving individual site information, should be investigated.
The following research gaps are identified (above): (a) climate–WT studies in the UK used a
limited number of WT sites or climate explanatory variables (focus on air temperature links to
WT) and /or are limited in geographical extent; (b) limited formal analysis of seasonality; (c)
limited knowledge of role of basin properties as modifiers of climate–WT associations; and
(d) need for alternative analysis method to optimise data utility.
Given this context, the aims of this study are (1) to improve the understanding of large-scale
spatial and temporal variability in climate–WT associations, and (2) to assess explicitly the
influence of basin properties as modifiers of these relationships. This paper resolves the issue
of driving data availability by using a comprehensive and consistent set of modelled climate
data (see Table 3 below). With a period of records of 1984–2007 (24 years), for a total of 35
sites located on 21 rivers within 16 basins (providing a Great Britain wide geographical
extent) six distinct modelled climatic variables were taken within 1 km of the sites. The study
focuses on broad spatio-temporal patterns; hence it is based on three-month averaged data
(i.e. seasonal). Such a temporal scale addresses issues of temporal auto-correlation often
found in water temperature time series (Caissie, 2006). The study also investigates a wider
range of basin properties than previous studies.
Innovatively, this paper investigates both site and national spatial scales at once. Multi-level
(ML) modelling with linear multiple regressions is applied as an alternative to site-specific or




to classification-based analyses because it allows pooling of all site data together while taking
into account data structure (i.e. observations at site, sites within same basin) as well as not
losing any information (Zuur et al., 2009). With this modelling technique, it is possible to
investigate both study aims (i.e. the broad climate-WT associations common to all sites, and
the site-specific responses which may be related to basin properties) within the same analysis
framework. In addition, model selection used Multi-Model Inference (MMI), another state-of-
the-art technique, which provides more robust models based on sets of good models rather
than selecting a single best model (Grueber et al., 2011).
**2   Data**
With regards to research Aim 1 of this paper, observed river temperature data were assembled
with a view to maximise spatial and temporal coverage as much as practically possible. To
address the issue of mismatching monitoring networks, climate variables were obtained from
a modelled dataset. The paired climate–WT dataset used in this paper has been published
online via an open-access data repository (Laizé and Bruna Meredith, 2015). With regards to
Aim 2, a comprehensive and consistent set of basin properties were derived for all study sites.
**2.1    Water temperature data**
WT data (unit: ºC) were collated from various research projects run by the UK's Centre for
Ecology and Hydrology (CEH). The period of record, temporal resolution, and recording
method of the individual datasets vary. These datasets totalled 41 sites, of which 35 were
retained after quality-control (e.g. removal of duplicates; see Fig. 3). As often the case, water
temperature was not the main focus of these projects: fish for the River Frome (1 site, 1991-
2009, 15-min logger; Welton et al., 1999), Great Ouse (1 site, 1989-1993, hourly logger), and
Tadnoll (2 sites, 2005-2006, 15-min logger; Edwards et al., 2009) studies; impact of forestry
on water quality for the Plynlimon catchment project (4 sites, 1984-2008, weekly manual
recording; Neal et al., 2010); acidification monitoring for the UK Acid Water Monitoring
Network (UKAWMN) project (10 sites, 1988-2008, monthly (not necessarily on same day)
manual recoding; Evans et al., 2008); hydrological and biogeochemical processes for the
LOwland CAtchment Research (LOCAR) project (17 sites, 2002-2011, 15-min logger;
Wheater et al., 2006). Because these original projects were focused on natural rivers, the
temperature data used herein may be considered as largely free from artificial influences (e.g.
no industrial use for cooling or heated effluent discharges).





## 2.2 Climate data

The Climate Hydrology and Ecology research Support System (CHESS) dataset features six climate variables (Table 3). CHESS is the forcing dataset for the Joint UK Land Environment Simulator model (JULES; Best et al., 2011). CHESS is a UK-wide 1-km grid dataset derived by downscaling the UK Meteorological Office Rainfall and Evaporation Calculation System (MORECS) 40-km grids (Hough and Jones, 1997), except for precipitation that is based on observed rain gauge data (Keller et al., 2006). For each 1-km cell, modelled daily time series of all variables are available for the period 1971–2007. The processes linked to AT, LWR, P, and SWR are given in the stream heat budget overview (see Introduction) and summarised in Table 3. Specific humidity (SH) gives a measure of evaporation potential (i.e. the more humidity, the less evaporation due to reduced vapour pressure gradients; e.g. Hannah et al., 2008). Wind speed (WS) captures the various effects of wind in increasing evaporation (cooling) and convective air-water exchanges (cooling or warming) Each CHESS cell was matched to the study temperature site(s) it contained.

## 2.3 Seasonal time series

Firstly, sub-daily water temperature data were averaged at a daily time step (Frome, Great Ouse, Tadnoll, LOCAR) while spot measurements (Plynlimon, UKAWMN) were assumed representative of the day on which they were taken, although it is worth keeping in mind that they are only representative of daylight conditions. Secondly, daily water temperature data were matched by date to the daily climate data. Thirdly, seasonal averages were computed from these daily data for all variables. Seasons were defined as: December–February (winter), March–May (spring), June–August (summer), and September–November (autumn). For winter, these seasonal data for year $y$ were based on data from December of year $y$-1 to February of year $y$ (e.g. for 1976, December 1975, January and February 1976). Lastly, five time series were derived from these data: one series per season at an annual time step (i.e. winter 2000, winter 2001, winter 2002, etc.), and one series with all seasons at a seasonal time step (i.e. autumn 2000, winter 2000, spring 2000, etc). These series and their related models are referred to as thereafter 'autumn', 'winter', 'spring', 'summer', and 'all seasons'.





## 2.4 Basin properties
Basin properties were derived from the UK Flood Estimation Handbook (FEH), the UK
'industry standard' for flood regionalisation studies, which includes 19 basin descriptors
(Bayliss, 1999). A selection of descriptors was used, which are listed with detailed definitions
in the Methods section. These descriptors relate to elevation, permeability, and size, which are
known to modify hydroclimatological links (Bower et al., 2004; Laizé and Hannah, 2010;
Garner et al., 2014).
## 3 Methods
This section describes the analytical methods used. Firstly, as stated in the introduction, the
Multi-level (ML) modelling technique was chosen as the core method because it allowed to
analyse the multiple-site data in terms of both overall climate–WT associations (linked to
research Aim 1) and site-specific responses (linked to research Aim 2; role of basins as
modifiers of those associations). Secondly, with regards to overall climate-WT associations,
ML model selection was done with Multi-Model Inference (MMI) to yield more robust
models than with standard single model selection, especially given the number of climate
predictors used. Lastly, any relation between site-specific climate-WT responses and basin
properties were tested formally using an analysis of variance (ANOVA).
The study work flow is summarised in Fig. 2: (a) WT observed data linked with (b) modelled
climate variables, then (c) all converted to seasonal (three-month) average series used within
(d) ML modelling / MMI framework producing (e) five output models (individual seasons
and all seasons; Aim 1), and (f) sets of basin properties (Aim 2).
## 3.1 Multi-level modelling
To take into account the hierarchical nature of the water temperature dataset (e.g. sites located
on the same river), ML modelling was used to build linear models with water temperature as
the predicted variable, and the six climate variables as explanatory variables. When analysing
multiple-site datasets, there are two common alternatives: (a) performing one regression for
individual sites, or (b) one regression on all sites pooled together. On the one hand, site-
specific regressions can make results highly uncertain (sites may have few data-points; fitting
numerous regressions is more prone to identify spurious relationships, ie Type II errors).
Thus, drawing out general patterns (e.g. variation between sites, effect of site characteristics)
can be difficult. On the other hand, full pooling of sites ignores the clustering of samples





within groups, which may hide important differences between sites and may cause problems
with statistical inference (e.g. violation of the assumption of independence between samples,
sites with large or small numbers of samples equally influencing the model outcome).
To overcome these issues, ML modelling allows for the pooling of data from different sites
while taking into account the hierarchical data structure. A ML model has two components,
which correspond to generic patterns (i.e. similar to a regression on fully-pooled data) and to
level-specific patterns. The generic patterns, which are described by the explanatory variables
as in a standard regression, are called the 'fixed component' or 'fixed effects' of the model.
The unexplained variation between levels (i.e. site-specific patterns here) is termed the
'random component' or 'random effects'. The random component captures the fact that levels
may respond differently to a given predictor.
In our analyses, a three-level data structure was applied: individual observations (level 1)
nested within monitoring sites (level 2) nested within river stretches (level 3). In addition, a
time variable was included as a predictor to take into account any linear trend in the time
series. To avoid instability issues when fitting models, the predictors were centred (i.e.
predictor values minus their mean).

## 3.2    Model selection with multi-model inference

Following standard ML modelling practice (e.g. Zuur et al., 2009), the model selection was
applied in two stages: (a) selection of the random component variables; (b) selection of the
fixed component variables.
First, with all predictors included in the fixed component, models with the various
combinations of predictors in the random component were ranked using Akaike's Information
Criterion (AIC; Akaike, 1974) that selects models offering the best compromise between fit
and predictor parsimony (AIC corrected for small-size datasets, AICc was used). Selection
was done for the four seasonal series as well as the 'all season' series. In each case, the
random component giving the lowest AICc was retained.
Secondly, with the random component selected, the fixed component model selection
followed the MMI approach, which selects sets of 'good' models rather a single 'best' one.
Using a traditional model selection technique, like stepwise regression, the model with the
best (i.e. the lowest) AICc would be selected. This presents two issues: (a) due to the





algorithms underlying these types of selection techniques, some model formulations may end
up not being tested thus causing a sub-optimal selection; (b) given models with similar AICc
values have similarly good performance, it is not statistically correct to keep the lowest AICc
model only as the best model and discard the others. MMI addresses these issues by selecting
sets of good models. In practice, all possible combinations using from one to six of the
climate predictors described above are fitted. The resulting models are ranked based their
AICc. All models within four points of the lowest AIC are selected (Zuur et al., 2009).
Grueber et al. (2011) cover the above points in details and give a very good example of such
an application of MMI in a natural sciences context.
Akaike weights (Burnham and Anderson, 2002) were calculated; weights are re-scaled AICc
scores, which give an indication of the relative importance of each model within a set. If only
one model was tested, the weight would be one. Models with similar AICc scores have
similar Akaike weights. Weights are used when reporting on MMI outputs. Then, following
recommended statistical usage, all models within four points of the lowest AICc were selected
(Zuur et al., 2009). Note that in some cases, there is only one model selected because its AICc
is lower by more than four points from the next second model in line, and it would have the
higher Akaike weight too.
With MMI, the role of each explanatory variable is assessed using its relative importance
(RI). For a given predictor, RI is calculated as the sum of the AICc weights of the models in
which that predictor is included. RI ranges from 0 (variable never included) to 1 (included in
all models). For example, results showed that the 'all seasons' model is based on two models
with AICc weights 0.74 and 0.26; the explanatory variable P is only included in the latter
model, hence its RI of 0.26, while the other five predictors are in both models and have a RI
of 1 (see Table 4 below). With MMI, RI is analogous conceptually to predictor significance,
assessed with p values, in a standard regression model.

## 3.3    Analysis of basin property influence

For those explanatory variables that were included in the random effects (i.e. different sites
can have different coefficients), any relation between site-specific coefficients and site basin
properties was investigated by using maps and scatter plots of coefficients against basin
properties, and by applying ANOVA to confirm observed patterns. For each coefficient and
basin property, ANOVA is comparing formally (a) a model assuming there is no difference in





coefficient between sites against (b) a model assuming the coefficient is function of the basin
property. A basin property is considered having significant influence on the WT–climate
variable relationship when the ANOVA $p$ value is <0.05. To quantify the influence of these
properties, either alone or combined, linear regressions of the site-specific coefficients were
fitted.
**4    Results**
The result section has three parts:
• Selection and performance of the five models (all seasons, winter, spring, summer,
autumn).
• Analysis of the fixed component of the five ML models to inform on climate-WT
associations (research Aim 1); results are split in three sub-sections (relative
importance of the predictors, form and strength of predictor-WT associations, relative
contributions of predictors to modelled WT).
• Analysis of the random component of the five ML models to inform on site-specific
climate-WT responses (for those predictors included as random effects), followed by
ANOVA to assess the role of basins as modifiers of the climate –WT associations.
(research Aim 2).
**4.1    Model selection and performance**
As described above, selecting the five ML models was done in two stages. First, with all
predictors included in the fixed component of the ML model, combinations of predictors as
random effects were tested, and the combination yielding the lowest AICc was retained. As a
result, the following variables were included as random effects (i.e. variables for which
different sites have different coefficients): all seasons = AT and SWR; winter = SH; summer
= P; autumn = SWR; no predictor was included for spring.
Second, all combination of the predictors in the fixed components were tested with MMI. The
number of models included in each final set as selected by MMI was: all seasons = 2; winter =
4; spring = 12; summer = 6; autumn = 14. With MMI, each set of models is summarised as an
'average model', for which a given variable coefficient is its average value over all models in
the set. The average model coefficients are presented in Table 4. Thereafter, if unqualified,
the term 'model' means the average model for a given set of selected models.





All models perform adequately (as evidenced by plots of fitted against observed water
temperature data in Fig. 4).

### 4.2 Relative influence of climate drivers

### 4.2.1 Relative importance of the predictors

Predictor RIs for all average models are given in Table 4. First, there is no predictor with a
zero RI for any average model. This means that all predictors are used in all or part of the sets
of selected individual models. Predictors can be ordered by decreasing importance: AT (RI=1
for all models); SWR (RI=1 for four models, and 0.64 for the summer one); WS (RI=1 for
two models, and 0.33-0.68 for others); SH (RI=1 for two models, 0.34-0.53 for others); P
(RI=1 for one model, 0.15-0.41 for others); LWR (RI=1 for one model, 0.13-0.25 for others).
Second, each model has its own set of most important predictors (with RI > 0.50 as a
threshold, i.e. predictor included in half of the selected individual models): all seasons, all
predictors except P; winter, AT, SWR, WS, and SH; spring, AT, SWR, and WS; summer, all
predictors; autumn, AT and SWR.

### 4.2.2 Form and strength of associations between climate predictors and water temperature

The section focuses on the fixed effect coefficients of the predictors (i.e. coefficients valid for
all sites). Predictors AT, SWR and SH have positive coefficients for all models (i.e. increases
of these predictors are associated with a consistent warming effect on water temperature).
Predictors LWR, WS, and P have positive or (mostly) negative coefficients (i.e. increases of
these predictors are associated with warning or cooling, depending on season; Table 4).
The strength of the association varies with season. Comparing the absolute value of the
seasonal coefficients for each variable (not between variables as they have different scales):
AT, lowest in winter, highest in autumn; SWR, lowest in autumn, highest in winter; LWR,
lowest in winter, highest in summer; WS, lowest in autumn, highest in summer; SH, lowest in
autumn, highest in winter; P, lowest in summer, highest in autumn.

### 4.2.3 Relative predictor contributions

By definition, the predictors may have different units and orders of magnitude. Their
coefficients cannot be compared directly to get an indication of their relative contribution to



WT predictions. Instead, WT predictions were generated for the whole period of record and
the percentage contributions of each predictor to the WT modelled values were calculated.
Boxplots of the percentage contributions for the six predictors and the five models are
featured on the left-hand side of Fig. 5 (for readability, outliers are not displayed). The thick
black central line corresponds to the median percentage contribution. The shorter the boxes
and whisker extents are, the more constant are predictor contributions to modelled WT, with
longer extents representing more variation. While, the boxplots inform about contribution
differences between models, plotting predictor contributions against modelled WT (right-hand
side of Fig. 5) shows that the contribution variability, for a given model, is in many cases
related to WT rather than random (i.e. some predictors are more or less influential depending
on thermal conditions).
AT is the main contributor except in winter (second to SH); its median contribution is around
12% for winter, and 30-35% for the other models. In all cases, AT contribution increases as
WT increases (AT has more influence at warmer WT).
SWR influence is quite constant for all models (medians ranging from +4.5% to 7.5%; up to a
maximum of +15.8% in winter) except autumn, for which it is very limited (median +0.13%).
Within each model, SWR contribution is fairly stable across the WT range but showing
slightly more variability for colder WT.
LWR is the second contributor for the 'all seasons' and the summer models. Its contribution is
negative except for spring, but in all cases, the contribution decreases as WT increases (i.e.
LWR has more influence on colder WT).
WS has a negative contribution for all models except autumn. WS is most influential for
colder WT (e.g. down to a minimum of -13.70% for all seasons model, -11.74% for summer);
its contribution decreases as WT becomes warmer (e.g. around -1% for most models). WS
contributions are more variable for colder WT (ie more scatter right-hand side plots; Fig. 5)
than for warmer WT. For autumn, WS has limited influence, with its contribution ranging
from +0.17% to +0.90%.
SH contribution is highest in winter (main contributor with median +27.20%) and for 'all
seasons', but otherwise limited for the other seasons (medians ranging +2.10% to +7.23%).
SH contributions are independent from WT.





P has limited influence with its contributions ranging from -1.13% (minimum, spring) to
+0.22% (maximum, winter). Its contributions show very little variability and no pattern in
relation to WT.
## 4.3    Role of basin properties
The site-specific coefficients were initially mapped against elevation and permeability to
explore basin modification of the WT–climate relationship, and any pattern linked to
easting/northing. While there was no clear easting/northing pattern, the maps showed
potential associations between coefficients and basin properties.
As explained above, a set of 19 catchment descriptors were derived for each site. Many basin
properties co-vary, often substantially, and they are best interpreted as groups of properties
('meta-properties') rather than on their own. Descriptor specifications (Bayliss, 1999), pair
plots, and correlation matrices were checked to identify likely groups of descriptors (for
example, all FEH rainfall descriptors capturing basin wetness). Then, ANOVA was run on
those descriptors to identify the ones significantly associated with the model site-specific
coefficients. Finally, the descriptors for each meta-property were checked to confirm they
have consistent associations (positive or negative) with each model predictor. Considering the
basin properties significantly associated with the site-specific coefficients only, one descriptor
was retained to represent each meta-property.
The following meta-properties and their corresponding FEH descriptors were thus selected:
- Elevation/wetness ('elevation' hereafter): as noted in Laizé and Hannah (2010), basin
elevation and wetness are very strongly correlated in the UK; the meta-property
'Elevation' is represented by the FEH descriptor ALTBAR (mean basin elevation
above sea level; m) and, for the winter model only, by PROPWET (proportion of time
basin soils are wet (%), based on soil moisture time series classified as wet/dry days;
highly correlated to rainfall);
- Size: AREA (basin area; $km^2$); using its natural log;
- Permeability: BFIHOST (Base Flow Index from Hydrology of Soil Type (HOST);
dimensionless); ranging from 0 (less permeable basin) to 1 (more permeable);
The 35 study sites are representative of a wide range of UK basin types in terms of the above
properties: (1) upland/lowland (ALTBAR approximately within 20-700 m and PROPWET




within 24-80%); (2) small and medium size (AREA ~0.5-415 km$^2$); (3)
impermeable/permeable (BFIHOST 0.24-0.92). In addition, the study sites feature
combinations of all three meta-properties.
Associations between meta-properties/descriptors and site-specific coefficients are showed in
Table 5. Note: no property was found to be associated with P coefficients in summer.
To quantify the influence of the properties, either alone, or combined, simple linear
regressions of the site-specific coefficients were fitted and ranked with AICc following the
MMI technique used above. Models are featured in Table 6. The best models are the ones
with the lowest AICc (displayed in bold characters); while all models featured are within four
AICc points, hence are considered equally good (Zuur et al., 2009). Depending on the site-
specific coefficient, the R2 range from 0.125 (autumn SWR) to 0.411 ('all seasons' AT). In
each case, a single regression (on BFIHOST or ALTBAR) is the best model AICc-wise,
although most of the multiple regressions are within 4 AICc points so equally valid models.
These meta-properties are themselves not independent in the UK: (i) high upland basins are
impermeable generally (permeable geology occurs in the lowlands); (ii) there are
comparatively more small basins at higher elevation. Results in Table 6 demonstrate this. For
the 'all seasons' AT coefficient models, single regressions on BFIHOST, ln(AREA), and
ALTBAR achieves a R2 of 0.370, 0.284, and 0.127, respectively, but the multiple regressions
with either two or all of them only achieve R2 within 0.381–0.411. The comparatively small
gain when adding several predictors is due to the three properties co-varying. Similar
comments can be made on the other models.

## 22  5   Discussion

This section has two parts:
• Discussion of the ML modelling fixed components (national-scale patterns of climate-
WT associations; research Aim 1); this includes outcomes of MMI, physical
interpretation of the models, and dependence between climate-WT association and
season/temperature.
• Discussion of the ML modelling random components (site-specific climate-WT
responses to assess their modification by basin properties; research Aim 2); identified
basin properties are first considered individually, then combined.





## 5.1 Influence of climate drivers

This section discusses results related to the fixed component of the ML models, which provide information on national-scale patterns (i.e. patterns valid for every sites used in the analysis). As explained above, these patterns would be analogue conceptually to those sought by using cluster analysis or fully-pooled regressions but without their shortcomings (e.g. loss of information, issues with dependent observations). The use of ML modelling adressed one of the limitation of empirical regression-based models, for which temperatures are predicted at specific sites only. Note: the four seasonal models are by definition related to the 'all seasons' model, since they are based on subsets of the same original dataset, so that seasonal patterns are not independent from the 'all seasons' patterns.

The six climate predictors investigated were identified as significant within the MMI framework (note: MMI applied to the selection of the fixed component part of the ML models only). Standard model selection techniques (e.g. stepwise) would have most likely excluded the predictors that are not retained in all models of the MMI selected model sets (i.e. predictors with lower RI values). In this regard, this study illustrated how MMI can be useful in picking the effect of secondary controls, otherwise masked by dominant primary drivers.

The models broadly make sense against known physical processes. In interpreting model results, it important to bear in mind that the aim of the study was to assess the relative empirical associations between WT and the set of climate drivers, therefore the models are not explicitly process-based. In addition, the climate variables are inter-related in some extent (e.g. P associated with more cloud cover, hence reduced SWR and greater SH), and the analysis is based on 3-month averaged data, which may cause some aspects of the physical processes to be lost by the averaging (e.g. distinction between variable like SWR, only contributing during daylight and others like LWR contributing continuously).

All models flag a close association between AT and WT. This finding is consistent with the literature: it is well documented that AT and WT are both influenced by similar climatic drivers (e.g. incoming radiation), and tend towards thermodynamic equilibrium (Caissie, 2006). Both variables consequently tend to co-vary positively, making AT a very useful predictor (as it has been widely demonstrated in the literature; e.g. Webb and Nobilis, 1997), although the association is partly causal only (Johnson, 2003). SWR (insolation from sun) is physically a positive input of energy; and it is appropriately captured in the models with positive coefficients. In this study, LWR is the downward component of longwave radiation (see Table 3). From an energy budget perspective, LWR therefore corresponds to a positive



flux toward the river water. Consequently, LWR contribution to WT should be positive.
Results (Table 4 and Fig. 5) show this is necessarily the case. LWR corresponds to radiation
diffused by clouds, so co-varies positively with cloud cover (in addition, a pairwise plot of the
study dataset shows that within a given season LWR inversely co-varies with SWR).
Therefore, the negative WT-LWR associations would most likely be due to LWR acting as a
proxy for processes driving colder water temperatures (e.g. cloud cover). SH represents the
mass of water vapour in moist air. The rate of evaporation at the water surface is directly
proportional to the SH gradient (the more humid the air, the lower the evaporation rate). All
models give a positive association between SH and WT. As SH increases, the evaporation rate
decreases, and consequently, cooling due energy loss as latent heat decreases as well. WS has
a cooling effect by increasing evaporation at the water surface, which would be captured by a
negative contribution to WT. In addition, WS plays a significant role in air–water energy
exchanges by increasing mixing, which would manifest as increased cooling or warming
depending on the AT-WT gradient. For all models but autumn, WS has an overall negative
contribution (cooling). For the autumn model, the variable RI and its percentage contribution
are both low, so the positive association has to be considered with caution. P have positive or
negative coefficients depending on model. When rainfall occurs, its temperature may be
higher or lower than that of the river depending on season. In addition, P can also act as a
proxy for cloud cover, thus for reduced SWR and increased LWR. P has limited importance
and percentage contribution in all the models, which is probably due to precipitations being
event-based whereas other variables are continuous (e.g. AT).
The form and strength of the climate-WT association vary depending on season and on WT
range, as showed by the variability in predictor coefficients and contributions. This most
likely captures that the dominating climate drivers and physical processes (e.g.
evaporation/condensation, radiative fluxes; see energy budget above) may change from one
season to another, or within the same season, from colder to warmer weather conditions. As a
consequence, the impact of short (e.g. seasonal climatic drought) and long term climate
variability or change, and of mitigation schemes (e.g. increasing riparian tree shading) on
stream temperature may not be uniform across time (e.g. higher long-term temperature
increases in winter and spring; Langan et al., 2001).
Most empirical models have been based on single AT-WT regressions (Caissie, 2006) with
very few using other climate predictors (e.g. AT and solar radiation; Jeppesen and Iversen,
1987). The present study demonstrated the potential of several other climate variables to





contribute explanatory power (even if they are weaker predictors than AT), which can be
beneficial when trying to tease out the relative influences of the various interconnected
processes controlling water temperature regimes, or when AT is not available at a site.
Although this was not the primary objective of the study, the models could be used to
generate seasonal water temperatures for the whole spatial and temporal extent of the CHESS
datasets (whole country, 1971–2007 period of records), for example allowing to investigate
broader geographical pattern, or the impact of extreme events like drought.

## 8   5.2    Role of basin properties

The analysis of the random component of the models (i.e. site-specific) identified
permeability, elevation, and basin size as the main modifiers of the climate-WT response
(note: unlike for the fixed component, the random predictors were selected using standard
AIC, i.e. there is only one random component formulation for each of the five models). The
use of ML modelling addressed the limitations of empirical regression-based models to work
across different spatial scales (see above; Caissie, 2006). The basin properties are first
reviewed individually, then together to assess how their respective influences may combine
within a basin (i.e. are all influences cumulating, or one property dominating?)
For all models and for all predictors (all seasons AT, autumn SWR, winter SH), the more
(less) permeable the basin, the lower (higher) the coefficients. Thus, water temperature in
impermeable basins appears to be more sensitive to climate than in permeable basins. Indeed,
in permeable basins, the temperature regime is comparatively more influenced by the
groundwater input to the river; groundwater temperature tends to have more inertia and to
have a damper effect on river WT (groundwater warmer than river in winter, cooler in
summer) - see for example, Webb and Zhang (1999), Hannah et al. (2004), Caissie, 2006,
Kelleher et al. (2012). This pattern is consistent with Garner et al. (2014), which used
different temperature monitoring sites and basin properties to investigate air–water
temperature associations only.
With regard to basin size, results can be summarised as follows: (a) 'all seasons' model, WT
in smaller basins is more sensitive to AT but less sensitive to SWR than in larger basins; (b)
autumn mode, WT in smaller basins is more sensitive to SWR; (c) winter model, WT in
smaller basins is more sensitive to SH. Although, there are seemingly contradictory patterns
for SWR, this can be explained by the modelling. Where studies typically use only one





variable to represent the whole climate (e.g. AT, Garner et al., 2014), several climate
predictors are considered herein. As noted in the Introduction, AT and SWR co-vary in some
extent. In the 'all seasons' model, AT and SWR were both selected to capture the between-
site variability of the climate-WT response, while in the autumn model, only SWR was
retained. As a consequence, in the autumn model, SWR represents climate control, most
probably capturing part of the WT variability explained by AT when both variables are
included as in the 'all seasons' model. Overall, WT is more sensitive to climate in smaller
basins. Then, the inclusion of both AT and SWR in 'all seasons' allows to refine the
assessment of river thermal sensitivity beyond climate as a whole, to different types of energy
processes: smaller streams are more sensitive to air-water heat exchanges but less sensitive to
radiative fluxes than larger streams. One can hypothesize that smaller streams have a lower
volume of water to heat up than larger streams but also are likely to experience greater
relative shading by riparian trees than wider rivers downstream.
This finding, at first, looks partly inconsistent with Garner et al. (2014), who concluded that
larger basins were more sensitive to climate than smaller ones, because (i) headwater stream
being located at the start of the network have less time than larger streams to reach
equilibrium with AT further downstream, and (ii) headwater streams are more likely to be
shaded (riparian woodlands, topography). However, Garner et al. (2014) was based on cluster
analysis; small basins were included in one cluster only, which also included permeable
basins. As a consequence, it is likely that permeability and size influences were in some
extent confounded. In contrast, the sites used in this paper cover all combinations of
size/permeability basin types. Secondly, as noted by Kelleher et al. (2012), within the small
stream type, one needs to distinguish between shaded (i.e. due to with riparian woodland or
topography) and exposed streams, with shaded streams behaving more like permeable
streams. Only basin-wide land cover information was available for 29 out of 35 sites: 27
basins are under 20% woodland. While one cannot exclude woodland being concentrated on
the riparian corridor of each site, it is sensible to assume the 35 sites have a mix of shaded and
exposed streams. Although it would explain the pattern with 'all seasons' SWR (more
shading, less incoming sun), the shaded headwater argument has to be considered
inconclusive in relation to the wider climate controls.
With regard to basin elevation, results can be summarised as follows: (i) 'all seasons' model,
WT in higher elevation basins is more sensitive to AT but less sensitive to SWR; (ii) winter





model, WT in higher elevation basins is more sensitive to SH. These patterns can be
explained partly by elevation, partly by the fact that permeability, size and elevation are not
strictly independent in the UK. As noted above, elevation and rainfall co-vary greatly in the
UK, so that upland basins are wetter than lowland basins, hence associated with greater
precipitation (i.e. with more cloud cover and consequently, less influenced by SWR). In terms
of basin types, the study sites have no upland permeable basins (the UK geology is such that
this type hardly occurs in any case), plus high elevation basins tend to be smaller basins. The
patterns observed with elevation, which are consistent with those for permeability and size,
are most likely partly reflecting the upland basins are also largely impermeable and smaller.
Although each property has been statistically identified as having an influence, the latter point
leads to investigating how these influences may combine. The regression models of site-
specific coefficients against permeability, size, and elevation presented in Table 6 provide
some quantification of the influence of basin properties, both on their own, and combined. In
each case, the best model uses a single basin property, although the retention of other
properties in the MMI sets confirms the role of all three. In three cases out of four ('all
seasons' AT, autumn SWR, winter SH), permeability (BFIHOST) is dominant. Therefore, the
patterns described above would be primarily set by basin permeability, then by size and
elevation. At one end of the spectrum, small, upland, and/or impermeable basins are the most
exposed to atmospheric heat exchanges, at the other end, large, lowland, and permeable
basins are the least exposed.

## 6   Conclusions

By focusing on a nation-wide set of water temperature sites and extensive climate dataset, this
study addressed some of the limits of previous UK papers (limited number of WT sites,
climate predictors, and /or geographical extent); it also investigated formally seasonal
patterns, and, by using a wide range of basin descriptors, improved knowledge of the role of
basin properties as modifiers of climate–WT associations.
With regards to the need to explore alternative modelling techniques to maximise data utility,
ML modelling allowed to model climate-WT responses both at site and at national scales,
thereby adressing the limitation of empirical regression-based models compared to
deterministic models (Caissie, 2006). In addition, the model selection based on the MMI
approach permitted to investigate climate variables that would been most likely excluded by
standard selection techniques, and identify their influence as secondary controls.





In relation to research Aim 1 (improved understanding of large-scale climate–WT
associations), the modelling exercise showed that all of the six climate predictors investigated
in this study play a role as a control of water temperature. AT and SWR are important for all
models/ seasons, while LWR, SH, and WS are important for some models/ seasons only. The
form and strength of the climate-stream temperature association vary depending on season
and on water temperature. The dominating climate drivers and physical processes may change
across seasons, and across the stream temperature range. The impact of climate variability or
change, whether short or long term (e.g. seasonal supra-seasonal, or inter-annual climatic
drought, long-term air temperature increaes), and the benefit of mitigation measures (e.g.
increasing shading) on stream temperatures need to be assesed accordingly.
In relation to research Aim 2 (assessing influence of basin properties as modifiers of climate-
WT associations), the study confirmed the role of basin permeability, size, and elevation as
modifiers of the climate-WT associations. The primary modifier is basin permeability, then
size and elevation. Smaller, upland, and/or impermeable basins are the ones most influenced
by atmospheric heat exchanges, while the larger, lowland and permeable basins are least
influenced (note: some basin types occur less frequently or hardly in the UK, e.g. upland
permeable). This means that, in addition to seasons and temperature range, the impact of
climate on stream temperatures and the benefits of mitigation schemes may vary with
location. This study shows the importance of accounting properly for the spatial and temporal
variability of climate-stream temperature associations and their modification by basin
properties.
**Data availability**
The dataset used in this paper is available from the NERC EIDC open-access data repository
(Laizé and Bruna Meredith, 2015).
**Acknowledgements**
The authors would like to thanks CEH colleagues for their help with (a) sourcing the water
temperature datasets, Mike Bowes, Francois Edwards, Ned Hewitt, Mike Hutchins, and
Gareth Old; and (b) retrieving the CHESS data, Eleanor Blyth, Douglas Clark, and Richard
Ellis. The authors acknowledge financial support from the Natural Environment Research
Council (NERC) through its National Capability funding to the Centre for Ecology and
Hydrology, and its PhD funding to the first author. Some of the material in this paper was
taken from the first author's PhD thesis.



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





1    Table 1. Climate–water temperature studies carried out in the UK.

| Reference | Number of Sites | Number of Basins | Location | Number of Climatic Variables | Length of Study Period |
|---|---|---|---|---|---|
| Wilby *et al.* (2014) | 36 | 2 | central England | 1 | 2 years |
| Garner *et al.* (2014) | 38 | 38 | England & Wales | 1 | 18 years |
| Broadmeadow *et al.* (2011) | 10 | 2 | south England | 3 | 3 years |
| Brown *et al.* (2010) | 6 | 1 | north England | 2 | 2 years |
| Hrachowitz *et al.* (2010) | 25 | 1 | northeast Scotland | 0 | 2 years |
| Hannah *et al.* (2008) | 2 | 1 | northeast Scotland | 7* | 2 years |
| Malcolm *et al.* (2004) | 6 | 1 | northeast Scotland | 1 | 3 years |
| Hannah *et al.* (2004) | 1 | 1 | northeast Scotland | 9* | 6 months |
| Webb *et al.* (2003) | 4 | 1 | southwest England | 1 | 5 years |
| Langan *et al.* (2001) | 1 | 1 | northeast Scotland | 1 | 30 years |
| Webb and Zhang (1999) | 2 | 2 | South England | 5 | 2 seasons |
| Evans *et al.* (1998) | 1 | 1 | west England | 9* | 17 days |
| Crisp (1997) | 5 | 1 | northwest Wales | 1 | 3 years |
| Webb and Zhang (1997) | 11 | 1 | southwest England | 4 | 2 seasons |

2    * includes different measurements of related climatic variables



1          Table 2. Water temperature datasets used in study.

| Dataset | Period of Record | Recording (Time Step) | Number of Sites |
|---|---|---|---|
| Frome | 1991-2009 | Logger (15 min) | 1 |
| Great Ouse | 1989-1993 | Logger (hour) | 1 |
| LOCAR | 2002-2011 | Logger (15 min) | 17 |
| Plynlimon | 1984-2008 | Manual (week*) | 4 |
| Tadnoll | 2005-2006 | Logger (15 min) | 2 |
| UKAMN | 1988-2008 | Manual (month*) | 10 |

2    *Approximately once a week or month but not necessarily on same day



1    Table 3. CHESS data.

| Climate Variable | Abbreviation | Units | Process |
|---|---|---|---|
| Air temperature | AT | $^{o}K$ | Convective energy exchanges at water surface; energy loss or gain |
| Long wave radiation | LWR | $W\ m^{-2}$ | Downward energy bounced back by clouds; energy gain |
| Specific humidity | SH | $kg\ kg^{-1}$ | Air moisture content; higher humidity reduces evaporation rate; energy loss (evaporation) or gain (condensation) |
| Precipitation | P | $kg\ m^{-2}d^{-1}$ $(mm\ d^{-1})$ | Advective exchanges; energy loss or gain |
| Short wave radiation | SWR | $W\ m^{-2}$ | Downward direct energy (i.e. insolation); energy gain |
| Wind speed | WS | $m\ s^{-1}$ | Increases evaporation (energy loss) and convective exchanges (air mixing; energy loss or gain) |





1    Table 4. Generic response for the five average models.

|  | all seasons | | winter | | spring | | summer | | autumn | |
|---|---|---|---|---|---|---|---|---|---|---|
|  | Coef. | *RI* | Coef. | *RI* | Coef. | *RI* | Coef. | *RI* | Coef. | *RI* |
| AT | 0.5824 | *1.00* | 0.3955 | *1.00* | 0.6815 | *1.00* | 0.4969 | *1.00* | 0.6860 | *1.00* |
| SWR | 0.0055 | *1.00* | 0.0193 | *1.00* | 0.0073 | *1.00* | 0.0049 | *0.64* | 0.0003 | *1.00* |
| LWR | -0.0149 | *1.00* | 0.0001 | *0.13* | 0.0020 | *0.18* | -0.0126 | *0.52* | -0.0013 | *0.25* |
| WS | -0.1348 | *1.00* | -0.0685 | *0.68* | -0.0774 | *0.63* | -0.3028 | *1.00* | 0.0181 | *0.33* |
| SH | 0.4664 | *1.00* | 0.6658 | *1.00* | 0.0772 | *0.34* | 0.1542 | *0.53* | 0.0507 | *0.37* |
| P | 0.0003 | *0.26* | 0.0007 | *0.15* | -0.0041 | *0.38* | -0.0004 | *1.00* | -0.0045 | *0.41* |



1 Table 5. Basin descriptors significantly related to site-specific model coefficients (ANOVA;

2 $p \leq 0.05$).

| Model | Predictor | Basin Meta-property | FEH Descriptor | Type of Association |
|---|---|---|---|---|
| all seasons | AT | Elevation | ALTBAR | Positive |
| | | Permeability | BFIHOST | Negative |
| | | Size | AREA* | Negative |
| all seasons | SWR | Elevation | ALTBAR | Negative |
| | | Size | AREA | Positive |
| autumn | SWR | Permeability | BFIHOST | Negative |
| | | Size | AREA* | Negative |
| winter | SH | Elevation | PROPWET | Positive |
| | | Permeability | BFIHOST | Negative |
| | | Size | AREA* | Negative |

3 *tested on natural log





1    Table 6. Linear regressions of site-specific coefficients as function of basin properties

2    (models ordered by increasing AICc; best model in bold characters, all other models are

3    within four AICc points of best model hence selected via MMI).

| WT Model | Coefficient | Linear Regression | $R^2$ | AICc |
|---|---|---|---|---|
| all seasons | AT | **BFIHOST** | **0.370** | **-31.3** |
| | | BFIHOST+ALTBAR | 0.403 | -30.1 |
| | | BFIHOST+ln(AREA) | 0.381 | -29.3 |
| | | BFIHOST+ln(AREA)+ALTBAR | 0.411 | -28.3 |
| all seasons | SWR | **ALTBAR** | **0.177** | **-277.5** |
| | | ALTBAR+ln(AREA) | 0.183 | -275.2 |
| | | ln(AREA) | 0.089 | -274.0 |
| autumn | SWR | **BFIHOST** | **0.125** | **-223.1** |
| | | ln(AREA) | 0.115 | -222.6 |
| | | BFIHOST+ln(AREA) | 0.136 | -220.9 |
| winter | SH | **BFIHOST** | **0.192** | **48.7** |
| | | ln(AREA) | 0.162 | 50.0 |
| | | BFIHOST+ln(AREA) | 0.203 | 50.8 |
| | | BFIHOST+PROPWET | 0.192 | 51.3 |
| | | PROPWET | 0.123 | 51.6 |
| | | PROPWET+ln(AREA) | 0.178 | 51.9 |



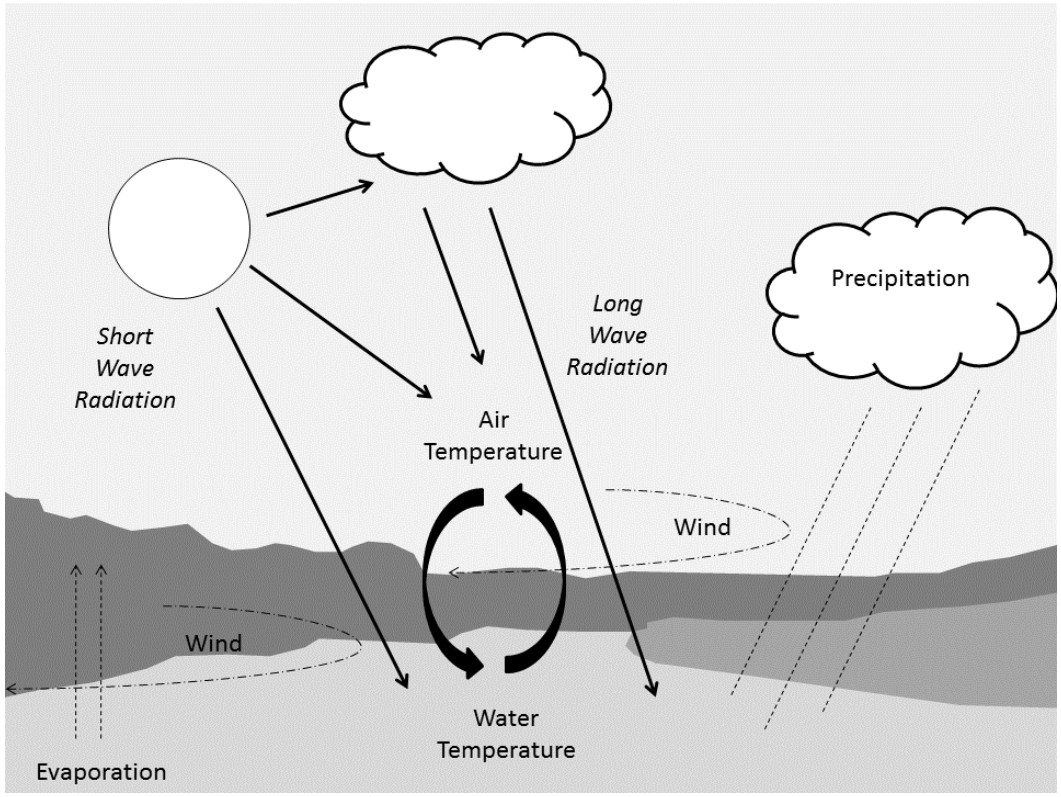

2    Figure 1. Multiple interdependent climate controls of water temperature [adapted from Caissie

3    (2006) and Hannah et al. (2008)].





Figure 2. Study flow chart**.**







Figure 3. Location map of the study sites.





2    Figure 4. Plots of observed and modelled water temperature for the five models.



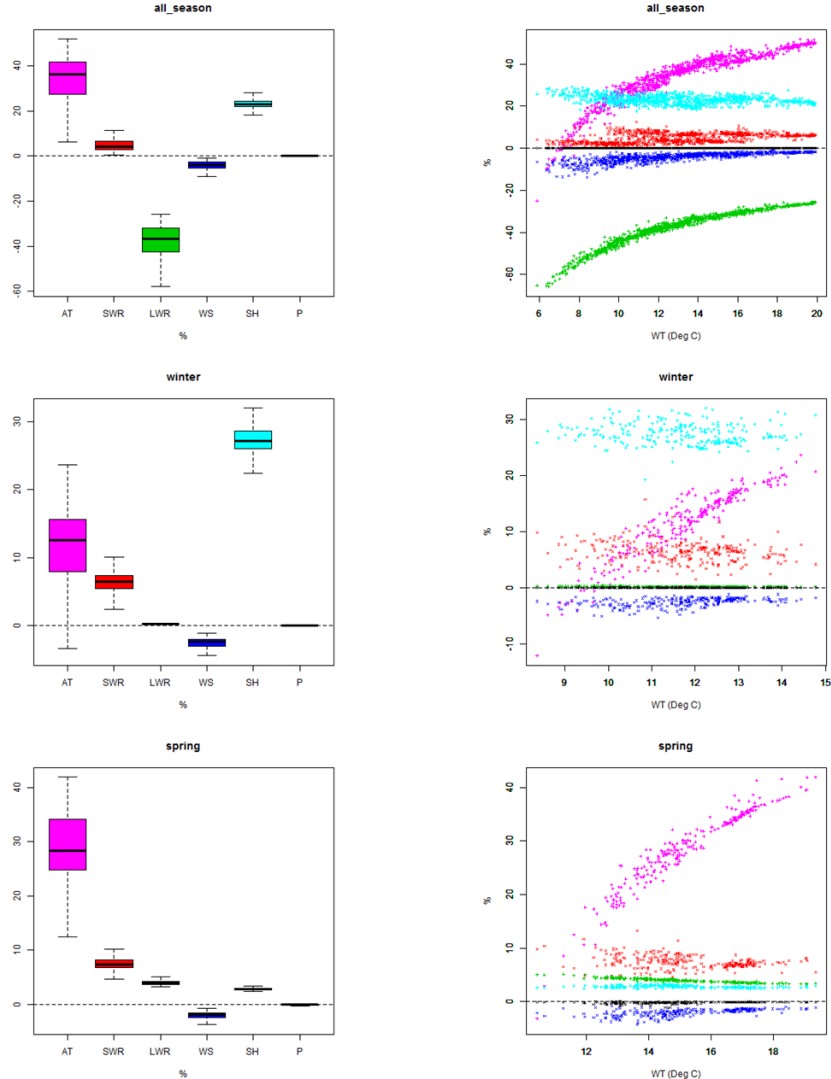

Figure 5a. Contributions of climate predictors to modelled WT (all seasons, winter, and
spring): left-hand side, boxplots of percentage contributions of climate predictors to modelled
WT values for all data-points (except outliers); right-hand side, scatter plots of percentage
contributions of climate predictors to modelled WT values against modelled WT values for all
data-points; colour-coding for all plots: magenta, AT; red, SWR; green, LWR; dark blue, WS;
cyan, SH; black, P.



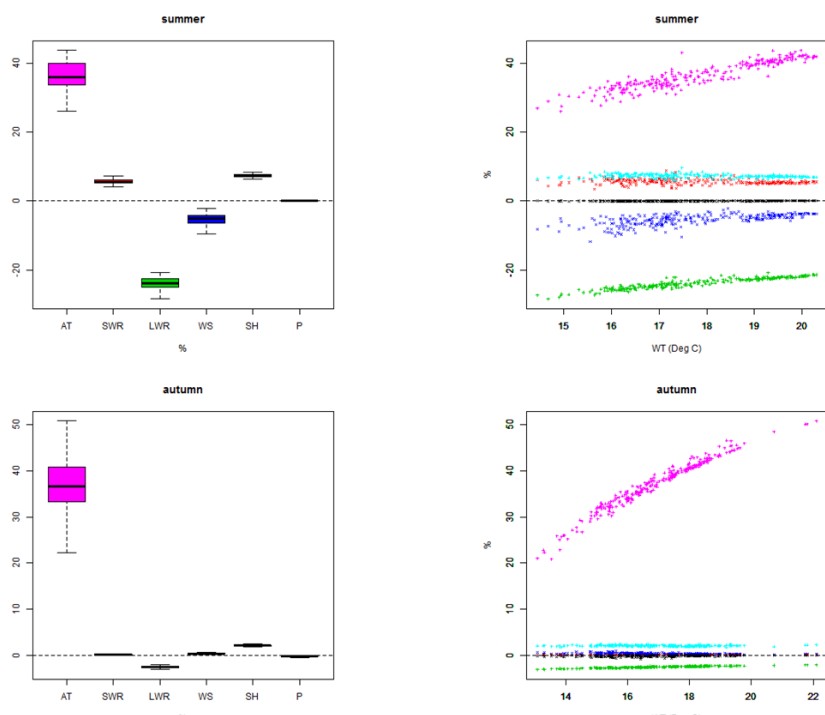

Figure 5b. Contributions of climate predictors to modelled WT (summer and autumn): left-
hand side, boxplots of percentage contributions of climate predictors to modelled WT values
for all data-points (except outliers); right-hand side, scatter plots of percentage contributions
of climate predictors to modelled WT values against modelled WT values for all data-points;
colour-coding for all plots: magenta, AT; red, SWR; green, LWR; dark blue, WS; cyan, SH;
black, P.