# Peer review of "Climate and basin drivers of seasonal river water"

_Hydrology and Earth System Sciences, 2016_

## Referee Comment (RC1) · Anonymous Referee #1 · 22 Jun 2016

This manuscript aims to develop a better understanding of the climatic controls on stream temperatures. Based on empirical models, long-term data from 35 sites across Britain are analysed. The objective is interesting and the paper in general worthwhile. I would have, however, appreciated a bit more clarity in the description of the modelling approach and somewhat more depth in the discussion of the results (e.g. what do the results imply? What can be learned from the results?). In places the manuscript comes across as rather sloppy (wrong numbering of tables and figures, missing explanations, incomplete referencing and very little information/discussion about the comparability of data from different sources). Please find below some detailed comments:

(1) P.2,l.26-27: irrelevant. Can be condensed.

(2) P.3,l.19-20: possible, but please be more specific and add some reasoning.

[Figure]

(3) P.3,l.21, figure 1: perhaps add the symbols from equation 1 to highlight more which process is related to which heat flux

(4) P.3,l.31: Hrachowitz et al. (2010) would also fit in nicely here.

(5) P.4,l.4-6: I found this a bit exaggerated. There are in fact quite some studies that consider a range of catchment properties (e.g. Isaak and Hubert, 2001; Scott et al., 2002; Moore, 2006; Nelitz et al., 2007; Hrachowitz et al., 2010; Isaak et al., 2010). Please tone down and add at least these references.

(6) P.4,l.25: table numbering is wrong. Table 2 is not referred to at all in the manuscript.

(7) P.4,l.29: "addresses" is unclear, maybe better to use "limits" or something similar

(8) P.5,l.20: figure numbering wrong: figure 3 referred to before figure 2. Please also make this figure a bit more informative. Provide basin/river names and potentially include elevation information. Please also clarify why some observation sites are far from streams (e.g. in insets 2 and 3).

(9) P.5,l.26: please provide more information on the actual data acquisition. Were the recorded values instantaneously measured temperatures or the averages over the logging intervals? How were the different sensors from the different studies placed and protected against radiative overheating? What about systematic uncertainties introduced by differential vegetation- and/or topographic shading at the different sites? Were the recorded data from the different studies pre-processed differently (e.g. filtering out overheating extremes)? What do different measurement precisions and accuracies of these different data sources imply for the analysis here? any systematic errors to be expected? And if not, why?

(10) P.6,l.6-7: Please be a bit more specific. How was precipitation regionalized based on rain gauge data? Kriging? IDW? Thiessen? Other methods?

(11) P.6,section 2.2: what about the uncertainties arising from the modelled climate data? How do they propagate through the temperature analysis here? Do they affect

the overall interpretation?

(12) P.6,l.20-22: what is the reasoning behind investigating seasonal averages? why only these? What is their ecological relevance? What about seasonal average daily (or 7-daily) maxima and minima? Would these not be more instructive? Just wondering.

(13) P.7,l.4-5: where is this section? I cannot find it. This is relevant information and needs to be shown.

(14) P.7,l.5: what is meant by "permeability"? permeability of what? How was it determined?

(15) P.7,l.6: not clear what is meant by this sentence

(16) P.7,l.23ff: how was the spatial correlation structure between sites along the same rivers accounted for? What was the flow distance between the sites closest to each other?

(17) P.7,l.24: it should at least be mentioned that linear models, in particular for the air-water temperature relationship, are oversimplifications and that for example logistic models can much better account for effects such as evaporative cooling (e.g. Mohseni et al., 1998)

(18) P.7ff, sections 3.1,3.2: I found this quite hard to follow. I would like to encourage the authors to invest some more effort to describe this critical part of their analysis more clearly.

(19) P.8,l.21-25: so how were the various combinations tested? Stepwise regression or best sub-set regression or some other method? What is AICc? How was it corrected for small-size datasets?

(20) P.10, section 4.1: not clear which explanatory variables were used in the individual models. All?

(21) P.11,l.1: what does "adequately" mean? Please provide R2 and p-values.

(22) P.11,l.3ff, section 4.2: one thing that is completely missing here but that may be of considerable relevance is the potential collinearity (or correlation) between the predictor variables, which can potentially result in highly unstable and misleading model results. It will therefore be necessary to quantify the collinearity and evaluate to which degree it actually influences the results.

(23) P.12,l.2: please clarify: are the percentage contributions in fact the proportions of the explained variance?

(24) P.12,l.4, figure 5: please provide a unit for the y-scale in the figure. The unit of the x-scale (%) seems to be wrong here. In addition, please be more specific: % of what?

(25) P.13,l.19ff: this needs to go into the methods section. Please also clarify why exactly these properties were chosen and provide a table with the relevant values.

(26) P.13,l.20: elevation not only related to wetness but clearly also to air temperature

(27) P.13,l.26: area is proxy for discharge and thus for thermal capacity, but is also linked to elevation

(28) P.13,l.27: what is the reasoning behind using HOST/permeability? What is it expected to explain?

(29) P.14,l.8: please also provide the individual p-values!

(30) P.14,l.14-15: this is a sweeping generalization which needs to be toned down

(31) P.14,l.16: why should there be more small basins at higher elevations? Channel formation does not have anything to do with elevation, but rather with contributing area and local slope. There may be some correlation with elevation but it is not generally valid as posed here. what, however, is true is that, necessarily the opposite is true: there are more larger basins at lower elevations.

(32) P.16,l.5-6: this is possible, but not sufficiently substantiated by data here. I would argue that it is equally likely the indirect correlation is merely a model artifact without

physical meaning (and potentially related to collinearity).

---

## Referee Comment (RC2) · Anonymous Referee #2 · 12 Nov 2016

This article explores basin and climatic drivers of stream temperatures across the UK. While the authors do a nice job throughout stating what is novel with respect to the study, I have a hard time finding some of their results novel. They show that air temperature, and solar radiation, drive heat fluxes throughout the year. Their findings fall in line with 30+ years of stream temperature research. A potentially novel result is the inclusion of different climatic factors, and the modeling style that they use to include these factors. However, it is not clear what this information adds to predictive capacity for stream temperature across the UK. Does including these variables mean there is greater explanatory power? Tertiarily, they also relate models to basin properties. However, the basin properties that were included are not well described in the paper, and end up feeling tangential to the other results. I'm left wondering where the model(s) perform(s) well, and where they performs poorly, and how performance changes across

different scales. Can this approach we used to improve modeling of stream temperatures? This is mentioned briefly at the end. As it stands, showing that models identify climatic variables as important seems to confirm what we already know. Showing, again, that basin properties influence these results is also potentially not new. I'm also left wondering about some of the implications of their data (in terms of temporal and spatial extent) for their conclusions. Overall, this is clearly a well-developed idea that will advance stream temperature research, but I am left feeling confused about broader implications, the sites in question, and whether this type of approach gets us any closer to improving our empirical modeling of stream temperatures.

Major comments: -Results, especially in tables and figures, are not presented in a way that enables easy interpretation by the reader. Table 6 means nothing to anyone but the authors. Table 5 – why is the FEH descriptor included, except for reference to Table 6? Why were the selection of descriptors used? Greater insight on which descriptors were included would be helpful. Section 4.3 for instance, refers to the abbreviations of FEH variables, but it would be much fewer words to just state the actual variables in text, and indicate FEH variables in parentheses

-The introductions to each section are not helpful, but I leave this up to the authors. I find that they detract from the reading of the manuscript.

-Sites with very different time scales of measurement where included. I get why this was done – there is not a lot of stream temperature data (a problem I am also having!). However, I'd like to know more about what is the effect? Were sites with 15-minute versus weekly and monthly data treated differently? With so many sites, it would be worth testing if 15-minute data were treated in the same way as weekly or monthly sites, what the effect on conclusions would be? If sites from weekly/monthly data were excluded, are conclusions different?

-Unclear what kind of variability in terms of basin/river properties your paper explores – a figure to this effect would be a good contribution. For instance, where else would

your results be comparable to? This would be helpful to know both in terms of stream temperature regime and basin properties.

-Magnitude of fluxes depend not only on climatic variables, but also on water temperature. Is model able to include this interaction, as it is a key determinant of evaporation/condensation and convection/conduction?

-Need more information on descriptors. They're included haphazardly. Don't even know which predictors are included in the model.

Minor comments: -Pg 3, Lines 10 – 20 – variables should have subscripts -Pg 3, Line 28 – misplaced comma -Pg 4, line 6 – consider the role of basin properties with respect to what? There's several papers in the US that have investigated the role basin properties may play in determining the stream temperature regime – they do so from an empirical perspective -Pg 5, line 3 – it's not clear to me what you mean by 'not losing any information' -Pg 6, section 2.2 – what impacts do you think using a 1km square meteorological dataset may have on your proposed conclusions? Are there any sites where microclimate could play a role? -Section 3.2 was difficult to follow and written confusingly. Comments were included in parentheses and not explained fully. The importance of AIC weights was introduced, but there was little explanation of what this value tells the reader (does 'relative importance' mean a better model? More trustworthy model?) -Some missing words in section 3.3 -Page 10 line 24: why was no predictor included for spring? -Abbreviations make the results difficult to adjust – I know what short wave radiation is, but every time I see SWR, I get confused! -Pg 17 line 1: Most other studies only use AT because it so well predicts stream temperatures. While your models demonstrate association, how much better do they predict stream temperature than air temperature alone? Furthermore, you use gridded AT data, which is available everywhere. I find it much less likely that AT is unavailable at a site with a suite of other climatic variables. -Pg 17, line 27 on – please rephrase out of list form -Figure 4 should be improved – it is difficult to read axes. Model fits should be included. -Figure 5, please label the y-axes

---

## Author Comment (AC1) · 9 Dec 2016

(1) P.2,l.26-27: irrelevant. Can be condensed.

Information requested by journal editor to acknowledge paper builds on first authors' PhD thesis. Sentence condensed as "This paper extends Laizé (2015)".

(2) P.3,l.19-20: possible, but please be more specific and add some reasoning.

The statement is backed up by a reference (Caissie, 2006).

(3) P.3,l.21, figure 1: perhaps add the symbols from equation 1 to highlight more which process is related to which heat flux.

The figure is being revised as suggested by the reviewer.

[Figure]

(4) P.3,l.31: Hrachowitz et al. (2010) would also fit in nicely here.

Citation added.

(5) P.4,l.4-6: I found this a bit exaggerated. There are in fact quite some studies that consider a range of catchment properties (e.g. Isaak and Hubert, 2001; Scott et al., 2002; Moore, 2006; Nelitz et al., 2007; Hrachowitz et al., 2010; Isaak et al., 2010). Please tone down and add at least these references.

The main point was in fact that they were very few studies in the UK (Hrachowitz et al. (2010) being one, and actually already cited in Table 2), and not that many, relatively speaking, internationally (suggested references are largely focusing on North America). Sentence (page 4 l 7-9) edited accordingly, with additional references (except for Scott et al. (2002) and Moore (2006), which we could not find based on the name and year only).

(6) P.4,l.25: table numbering is wrong. Table 2 is not referred to at all in the manuscript.

It seems there was a technical glitch when preparing the manuscript for upload. Indeed, current Table 2 was marked for deletion so that current Table 3 should have been Table 2, etc. We corrected the manuscript by deleting Table 2 and updating table numbers and references accordingly.

(7) P.4,l.29: "addresses" is unclear, maybe better to use "limits" or something similar

Text changed as suggested above.

(8) P.5,l.20: figure numbering wrong: figure 3 referred to before figure 2. Please also make this figure a bit more informative. Provide basin/river names and potentially include elevation information. Please also clarify why some observation sites are far from streams (e.g. in insets 2 and 3).

Current Fig 3 was meant to be Fig 2, and vice-versa, so was correctly referred to first. We swapped figures 2 and 3, and corrected numbering accordingly. All sites are on

streams, but we only had access to a simplified river shapefile, which does not show smaller streams. A similar map was made with elevation as a background so could be provided as a replacement. Obtaining a more detailed river network to improve the inset is likely to take a significant amount of time.

(9) P.5,l.26: please provide more information on the actual data acquisition. Were the recorded values instantaneously measured temperatures or the averages over the logging intervals? How were the different sensors from the different studies placed and protected against radiative overheating? What about systematic uncertainties introduced by differential vegetation- and/or topographic shading at the different sites? Were the recorded data from the different studies pre-processed differently (e.g. filtering out overheating extremes)? What do different measurement precisions and accuracies of these different data sources imply for the analysis here? any systematic errors to be expected? And if not, why?

In Section 2.1, we cited the peer-reviewed papers related to the original datasets and covering the data acquisition. We also gave summary information. We feel that giving further details would require too much space. However, we clarified that fact that measurements are instantaneous whether they are manual or via a logger (l29, page 5). Regarding systematic differences between sites due to different recording processes, site characteristics, etc. (which are indeed to be expected), this was the main reason to use multi-level models. Multi-level models are models that take into account data structure; for example, if you had 2 sites, one shaded, one not, the regression slope and intercept for each site would be different to reflect that one site is, let's say, cooler on average and more responsive than the other.

(10) P.6,l.6-7: Please be a bit more specific. How was precipitation regionalized based on rain gauge data? Kriging? IDW? Thiessen? Other methods?

The text has been edited to clarify that precipitation data were derived from observed rain gauge data by using the natural neigbour interpolation method, which is a devel-

opment of the Thiessen approach.

(11) P.6,section 2.2: what about the uncertainties arising from the modelled climate data? How do they propagate through the temperature analysis here? Do they affect the overall interpretation?

The climate data are in fact deterministic (one set of climate data); some of the variables are in fact interpolated based on observations (eg precipitation), and we fitted one time series with other time series. In this sense, we did not analyse uncertainty as one may do with GCM outputs generating ensemble runs of several thousands. If one think in terms of how good CHESS data represent climate variables, we checked with our colleagues and they are of the opinion that the main weakness in the CHESS data was the downscaling of MORECS data from 40km to 1km, which may cause some variables to be overestimated in some parts of the UK; however, we had no sites located in those parts. Given the models performed reasonably well at predicting the observed water temperatures (conditional R-squared obetween 0.84-0.96), we consider that any uncertainty is acceptable and does not affect the overall analysis massively. In addition, with multi-level modelling, confidence intervals, although they can be calculated, are not considered as meaningful as for standard regression models.

(12) P.6,l.20-22: what is the reasoning behind investigating seasonal averages? Why only these? What is their ecological relevance? What about seasonal average daily (or 7-daily) maxima and minima? Would these not be more instructive? Just wondering.

Ecological relevance is with regards to phenology. A clarification on this was added in the introduction, page 3, l31. In addition, research fitted within a wider research on seasonal hydro-climatic patterns (eg Laize & Hannah, 2010). Minima and maxima would be of interest if investigating topics like lethal thresholds.

(13) P.7,l.4-5: where is this section? I cannot find it. This is relevant information and needs to be shown.

This information is actually in the Results section, not Methods (text was corrected). See comment 25 below; reviewer suggested this should move to the Methods section.

(14) P.7,l.5: what is meant by "permeability"? permeability of what? How was it determined? Catchment permeability in the sense of flashy impermeable catchments vs groundwater-fed catchments.

We added a clarification in the text. It is characterised by using catchment base flow index (BFI; described later in the text).

(15) P.7,l.6: not clear what is meant by this sentence

These basin properties are generally recognised in UK studies (those cited and many others) as modifiers of climate-hydrology associations. Sentence rephrased for more clarity.

(16) P.7,l.23ff: how was the spatial correlation structure between sites along the same rivers accounted for? What was the flow distance between the sites closest to each other?

It was taken into account by using multi-level modelling. As explained in the method section, the multi-level models were specified with 3 levels: data, data at a site, sites on a river. With the level representing rivers, the multi-level models were able to take into account the fact that two sites on the same river may have more similar records than two sites on different rivers due to their physical linking.

(17) P.7,l.24: it should at least be mentioned that linear models, in particular for the air-water temperature relationship, are oversimplifications and that for example logistic models can much better account for effects such as evaporative cooling (e.g. Mohseni et al., 1998).

We added this point in section 3 methods (and reference to Mohseni et al., 1998).

(18) P.7ff, sections 3.1, 3.2: I found this quite hard to follow. I would like to encourage

the authors to invest some more effort to describe this critical part of their analysis more clearly.

We appreciate the methods may be difficult to understand. We reviewed these sections attempting to clarify what we think may be the more confusing points (the reviewer's comment is not specific in this regards).

(19) P.8,l.21-25: so how were the various combinations tested? Stepwise regression or best sub-set regression or some other method? What is AICc? How was it corrected for small-size datasets?

As described, all combinations were fitted (programmatically) and their AICs calculated. No stepwise or subsetting involved. The model with the lowest AICc was retained. Text was slightly edited to clarify the process. AICc is one standard option in R; the difference with AIC is a slightly modified formula putting more penalty on the number of parameters than with normal AIC. This is the suitable thing to do in this study (there are accepted rules about number of predictors v sample size). To the non specialist, this is actually a minor technical detail, which has been included for completeness. Detailing AICc any further would require to detail AIC, which is itself fully described in Akaike (1974); AIC and AICc are nowadays standard tools of the trade.

(20) P.10, section 4.1: not clear which explanatory variables were used in the individual models. All?

Yes, some predictors were used in all models (RI = 1), some predictors were used in some of the models constuting a set of best models (RI < 1).

(21) P.11,l.1: what does "adequately" mean? Please provide R2 and p-values. When using multi-level modelling (and moreover within the multi-model inference framework), R2 as commonly featured for regressions are actually not suitable, hence the current choice of showing observed vs predicted plots only. However, to address the reviewer's point, there is an alternative R2 for multi-level models ("conditional R2") by Nakagawa

and Schielzeth (2013), which we calculated and added to section 4.1 (reference to Nakagawa paper added too). Word "adequately" removed. Regarding p values, their conceptual equivalent within a MMI framework is the predictor relative importance RI used in the paper. A sentence in the method section 3.2 (l 25-26, page 9) and one in section 4.2.1 (l15-17, page 11) have been added to highlight this, and clarify that higher RI means more significant predictors.

(22) P.11,l.3ff, section 4.2: one thing that is completely missing here but that may be of considerable relevance is the potential collinearity (or correlation) between the predictor variables, which can potentially result in highly unstable and misleading model results. It will therefore be necessary to quantify the collinearity and evaluate to which degree it actually influences the results.

We were aware of possible collinearity issues and this was one of the reasons to use MMI. Collinearity gives inflated standard errors of parameter estimates. Approaches like MMI are fairly robust to even high levels of collinearity (see for example, Feckleton, 2011; Grueber et al, 2011); simply put, if there is high collinearity between two variables, then they don't appear in the same model, and don't force standard errors up. In addition, in this case, correlations between predictors were for most pairs below 0.5 (Pearson), and even for the more highly correlated pairs still within what is generally regarded as reasonable by statisticians. In the results/discussion, we aimed to take into account the implications of predictors co-varying when interpreting observed patterns.

(23) P.12,l.2: please clarify: are the percentage contributions in fact the proportions of the explained variance?

Sentence was slightly edited to clarify: for each record in the dataset, WT was predicted using the average model coefficients from Table 3, then the % contribution of each predictor to predited WT calculated (ie we made a time series of WT predictions and predictor % contributions).

(24) P.12,l.4, figure 5: please provide a unit for the y-scale in the figure. The unit of the

x-scale (%) seems to be wrong here. In addition, please be more specific: % of what?

The y-axis label ('%') was erroneously placed on the x-axis; figure amended. Captions explain what the % are for both sets of plots (we found that trying to abbreviate the % definition to fit it as an axis label did not really improve the figures).

(25) P.13,l.19ff: this needs to go into the methods section. Please also clarify why exactly these properties were chosen and provide a table with the relevant values.

The section on basin properties was originally in Methods but, because part of the analysis was used to confirm the selection of FEH properties (out of the available 19 properties) and because the Methods section was already quite long, we felt that it could be considered part of Results instead. We propose to move this back to Methods, where readers would more likely expect it. Regarding values, we assume the reviewer means the actual property values per site: we have originally included ranges of values only to save space (section 4.3) but we could include a table, perhaps as an appendix or as supplemental material. See also response to Reviewer #2 comments below.

(26) P.13,l.20: elevation not only related to wetness but clearly also to air temperature

We added this comment when elevation is introduced.

(27) P.13,l.26: area is proxy for discharge and thus for thermal capacity, but is also linked to elevation

We added this when the property is introduced. Note that this is also already mentioned in the Discussion.

(28) P.13,l.27: what is the reasoning behind using HOST/permeability? What is it expected to explain?

We were expecting groundwater-fed catchments to behave differently from impermeable ones (eg temperature regime influenced by groundwater inputs). We added this when the property is introduced. It is also covered in the Discussion.

[Figure]

(29) P.14,l.8: please also provide the individual p-values!

Please see response to similar comments re using MMI and selection with an information criterion (ie p values are not relevant). The models here were selected using MMI as per the main WT models. But, unlike for the main models, for which only the average model was featured, Table 6 lists all the models (and their R-squared and AIC) included in a MMI model set. This may give the impression the models were fitted using traditional approaches (eg removing predictors with high p values as not significant) although it was not the case.

(30) P.14,l.14-15: this is a sweeping generalization which needs to be toned down

Sentence revised.

(31) P.14,l.16: why should there be more small basins at higher elevations? Channel formation does not have anything to do with elevation, but rather with contributing area and local slope. There may be some correlation with elevation but it is not generally valid as posed here. what, however, is true is that, necessarily the opposite is true: there are more larger basins at lower elevations.

Sentence changed as suggested.

(32) P.16,l.5-6: this is possible, but not sufficiently substantiated by data here. I would argue that it is equally likely the indirect correlation is merely a model artifact without physical meaning (and potentially related to collinearity).

An early version of the manuscript actually made that very point. We re-instated it but kept the possible physical explanation as well.

---

## Author Comment (AC2) · 9 Dec 2016

It is worth clarifying that we did not aim to produce a better predictive model of water temperature, but rather, the modelling exercise was a mean to gain better understanding of the large-scale spatial and temporal variability in climate–WT associations, and of the influence of basin properties on these associations. The modelling techniques (multi-level modelling and multi-model inference) are definitively novel in their application to water temperature. In particular, we could analyse data both from at site scale and at national scale at once with multi-level modelling. The sites covered a reasonably wide range of catchment types. The combined wider spatial patterns and site-specific responses related to basin properties help unraveling the relative influence of climate vs land surface control across scales.

[Figure]

(1) Results, especially in tables and figures, are not presented in a way that enables easy interpretation by the reader. Table 6 means nothing to anyone but the authors. Table 5 – why is the FEH descriptor included, except for reference to Table 6? Why were the selection of descriptors used? Greater insight on which descriptors were included would be helpful. Section 4.3 for instance, refers to the abbreviations of FEH variables, but it would be much fewer words to just state the actual variables in text, and indicate FEH variables in parentheses

We think this comment, and two others below, as well as some comments from Reviewer #1 stem from the way we introduced the basin properties. As we stated in response to Reviewer #1 comment 25, we would move the bulk of Section 4.3 back to Methods, thus streamlining and clarifying which basin descriptors we used. Re Table 5, we included the FEH descriptor to highlight the fact that the basin property (eg elevation) is characterised by a specific descriptor (eg ALTBAR); many different descriptors could be used to characterise elevation. Then, since these descriptors are indeed used for the results featured in Table 6, we thought it would help readers to make the connection. We believe we explain Table 6 clearly enough in the text. Regarding the latter point of FEH descriptors abbreviations, they are actually not abbreviations as such but short names (except for BFIHOST), so their explicit names is already given (eg ALTBAR = mean basin elevation above sea level). To clarify things, we swapped order of full name and FEH short name.

(2) The introductions to each section are not helpful, but I leave this up to the authors. I find that they detract from the reading of the manuscript.

Experience with past papers showed that some readers benefit from these section introductions. We are inclined to keep them unless there is a need to reduce the length of the manuscript.

(3) Sites with very different time scales of measurement where included. I get why this was done – there is not a lot of stream temperature data (a problem I am also having!).

[Figure]

However, I'd like to know more about what is the effect? Were sites with 15-minute versus weekly and monthly data treated differently? With so many sites, it would be worth testing if 15-minute data were treated in the same way as weekly or monthly sites, what the effect on conclusions would be? If sites from weekly/monthly data were excluded, are conclusions different?

The discrepancy in data time scales is handled by using multi-level modelling. So, for example, if sites based on 15-min data behaved differently from sites based on weekly data, the model would correct for that. However, it would not explicitly investigate what the effect of one over the other. We added a mention of this as possible future research.

(4) Unclear what kind of variability in terms of basin/river properties your paper explores – a figure to this effect would be a good contribution. For instance, where else would your results be comparable to? This would be helpful to know both in terms of stream temperature regime and basin properties.

In Section 4.3, we have included a paragraph giving the ranges of basin properties for the 35 sites. They do provide a fairly wide range of basin types in the UK. The original data sources include a mix of lowland permeable (eg from LOCAR, Tadnoll; the UK has most of its aquifers in lowland regions) and impermeable basins, upland impermeable (eg form Plynlimon), as well as small to medium basin sizes. Sites from the AWMN cover all types. The gap in coverage may be in terms of large basins (ie >1000 km2) but these are far less common in the UK than in other countries. If needed, we could provide the detail properties per site (table) to be included as appendix or supplemental material.

(5) Magnitude of fluxes depend not only on climatic variables, but also on water temperature. Is model able to include this interaction, as it is a key determinant of evaporation/condensation and convection/conduction?

As it stands, the models are at their core linear regression with water temperature as the dependent variable. Feedbacks cannot be built-in. This would require a different

type of method, possibly bespoke models handling iterative calculations. However, we explored this with the outputs from Fig 5 showing how the contributions of different predictors change with water temperature.

(6) Need more information on descriptors. They're included haphazardly. Don't even know which predictors are included in the model.

See previous responses to comments (also to Reviewer #1) about Section 4.3 (we believe the proposed changes to the way basin properties are presented will address this).

(7) Pg 3, Lines 10 – 20 – variables should have subscripts

Text amended as suggested.

(8) Pg 3, Line 28 – misplaced comma

Comma moved to its proper position.

(9) Pg 4, line 6 – consider the role of basin properties with respect to what? There's several papers in the US that have investigated the role basin properties may play in determining the stream temperature regime – they do so from an empirical perspective

Sentence amended page 4 l 7-9 (also response to see Reviewer #1 comment #5).

(10) Pg 5, line 3 – it's not clear to me what you mean by 'not losing any information'

This refers to the loss of information due to class-level averaging with classification-based analyses (already covered in more detailed a few paragraphs before on page 4). We inserted a reminder/clarification page 5 line 5.

(11) Pg 6, section 2.2 – what impacts do you think using a 1km square meteorological dataset may have on your proposed conclusions? Are there any sites where microclimate could play a role?

We agree there may be micro-climate effects (one co-author published several papers

based on field site monitoring showing the impact of shading, etc.), but the focus of this study was the wider spatial patterns, which is quite novel, so we did not investigate this further. In addition, based on the information we had, we do not think there was any site where micro-climate effect was conclusively present. Obvious. However, we mention this in the discussion (for example, highlighting how shaded river may behave differently from non-shaded). We propose to add a mention of this point in the conclusion.

(12) Section 3.2 was difficult to follow and written confusingly. Comments were included in parentheses and not explained fully. The importance of AIC weights was introduced, but there was little explanation of what this value tells the reader (does 'relative importance' mean a better model? More trustworthy model?)

This particular section has been revised and streamlined to provide more clarity (see reviewer #1 comment #18).

(13) Some missing words in section 3.3

Section has checked and revised.

(14) Page 10 line 24: why was no predictor included for spring?

As explained in that paragraph, for spring, the model getting the lowest AICc (the best model) was the model with random intercept only (ie the only difference between sites is with regards to the mean water temperature; all sites have the same response slope for all predictors; hence "no predictor was included" in the random effects). We added a clarification that spring is a random intercept only ML model (page 10, l30). We also expanded section 3.1 (page 8, l14-19) to give more clarity regarding multi-level modelling, which in turns would help readers to understand what is meant in this paragraph.

(15) Abbreviations make the results difficult to adjust – I know what short wave radiation is, but every time I see SWR, I get confused!

We appreciate this problem, but we had to use abbreviation for the sake of conciseness (there are many references to the model predictors in the text), and following that,

consistency forces us to use the abbreviations all through out.

(16) Pg 17 line 1: Most other studies only use AT because it so well predicts stream temperatures. While your models demonstrate association, how much better do they predict stream temperature than air temperature alone? Furthermore, you use gridded AT data, which is available everywhere. I find it much less likely that AT is unavailable at a site with a suite of other climatic variables.

As stated, our objective was not to build a better predictive model but to understand the various climate-stream temperature associations. As such, we do not claim that these particular models do necessarily better than AT-WT models (and did not investigate this), but that using other climate predictors in addition to AT could be informative in some cases. We added a few words to avoid any misunderstanding on that point. Regarding the comment about AT being not available, it is true that if one uses gridded climate data, then AT is probably more likely to be available than some of the other variables: what we had in mind were field sites where air temperature measurements may be missing. This was a minor point however, so we deleted it from the sentence.

(17) Pg 17, line 27 on – please rephrase out of list form

Done.

(18) Figure 4 should be improved – it is difficult to read axes. Model fits should be included.

The size of the figure has been increased slightly to improve readability (figure has also been edited to amend one label that was not displayed properly). Model fits (conditional R2) have been added to text; given the conditional R2 are strictly speaking calculated for each model in a model set rather than for the average model (ie the average of all models in a model set), we think it is more accurate to leave the reader to gauge the average model fits visually than to add a mean R2 to the plots (see Reviewer #1 comment #21).

[Figure]

(19) Figure 5, please label the y-axes

The y-axis label ('%') was erroneously placed on the x-axis; figure amended.

―――――――――――――――――――

---

## Author Response (AR1)

Authors' response (Comment, Response, Change); page and line numbers refer to the marked-up revised manuscript.

**Response to Reviewer #1**

Comment #1: P.2,l.26-27: irrelevant. Can be condensed.

Response #1: Information requested by the journal editor to acknowledge that the paper builds on the first author's PhD thesis.

Change #1: Sentence condensed as "This paper extends Laizé (2015)".

Comment #2: P.3,l.19-20: possible, but please be more specific and add some reasoning.

Response #2: The statement is backed up by a reference (Caissie, 2006).

Change #2: None.

Comment #3: P.3,l.21, figure 1: perhaps add the symbols from equation 1 to highlight more which process is related to which heat flux.

Response #3: Figure revised as suggested.

Change #3: New figure inserted and caption edited accordingly.

Comment #4: P.3,l.31: Hrachowitz et al. (2010) would also fit in nicely here.

Response #4: Agreed.

Change #4: Citation added (P3, L32 in revised manuscript with Track Changes).

Comment #5: P.4,l.4-6: I found this a bit exaggerated. There are in fact quite some studies that consider a range of catchment properties (e.g. Isaak and Hubert, 2001; Scott et al., 2002; Moore, 2006; Nelitz et al., 2007; Hrachowitz et al., 2010; Isaak et al., 2010). Please tone down and add at least these references.

Response #5: The main point was in fact that they were very few studies in the UK (Hrachowitz et al. (2010) being one, and actually already cited in Table 2), and not that many, relatively speaking, internationally (suggested references are largely focusing on North America).

Change #5: The sentence (P4, L5-9 in revised manuscript) was edited accordingly, with suggested additional references added, except for Scott et al. (2002) and Moore (2006), which we could not find based on the name and year only.

Comment #6: P.4,l.25: table numbering is wrong. Table 2 is not referred to at all in the manuscript.

Response #6: It seems that there was a technical glitch when preparing the manuscript for uploading. Indeed, current Table 2 was marked for deletion so that current Table 3 should have been Table 2, etc.

Change #6: We corrected the manuscript by deleting Table 2 and updating table numbers and references accordingly.

Comment #7: P.4,l.29: "addresses" is unclear, maybe better to use "limits" or something similar.

Response #7: Agreed.

Change #7: Text changed as suggested above (P4, L32 in revised manuscript).

Comment #8: P.5,l.20: figure numbering wrong: figure 3 referred to before figure 2. Please also make this figure a bit more informative. Provide basin/river names and potentially include elevation information. Please also clarify why some observation sites are far from streams (e.g. in insets 2 and 3).

Response #8: Current Fig 3 was meant to be Fig 2, and vice-versa, so was correctly referred to first. We swapped figures 2 and 3, and corrected numbering accordingly. All sites are on streams, but we only had access to a simplified river shapefile, which did not show smaller streams. Smaller stream have been added, as well as elevation ranges as a background. Where available, river names were added.

Change #8: Updated Fig. 2, and corresponding caption.

Comment #9: P.5,l.26: please provide more information on the actual data acquisition. Were the recorded values instantaneously measured temperatures or the averages over the logging intervals? How were the different sensors from the different studies placed and protected against radiative overheating? What about systematic uncertainties introduced by differential vegetation- and/or topographic shading at the different sites? Were the recorded data from the different studies pre-processed differently (e.g. filtering out overheating extremes)? What do different measurement precisions and accuracies of these different data sources imply for the analysis here? any systematic errors to be expected? And if not, why?

Response #9: In Section 2.1, we cited the peer-reviewed papers related to the original datasets and covering the data acquisition. We also gave summary information. We feel that giving further details would require too much space. However, we clarified that fact that measurements are instantaneous whether they are manual or via a logger. Regarding systematic differences between sites due to different recording processes, site characteristics, etc., which are indeed to be expected, this was the main reason to use multi-level models. Multi-level models are models that take into account data structure; for example, if you had two sites, one shaded, one not, the regression slope and intercept for each site would be different to reflect that one site is, for example, on average cooler, or one site is more responsive to direct sunlight than than the other.

Change #9: We added clarifications (measurements are instantaneous; P6, L3-4 in revised manuscript).

Comment #10: P.6,l.6-7: Please be a bit more specific. How was precipitation regionalized based on rain gauge data? Kriging? IDW? Thiessen? Other methods?

Response #10: We clarified that precipitation data were derived from observed rain gauge data by using the natural neighbour interpolation method, which is a development of the Thiessen approach.

Change #10: The text has been edited accordingly (P6, L12-14 in revised manuscript).

Comment #11 P.6,section 2.2: what about the uncertainties arising from the modelled climate data? How do they propagate through the temperature analysis here? Do they affect the overall interpretation?

Response #11: The climate data are in fact deterministic (one set of climate data), some of the variables are interpolated based on observations (eg precipitation), and we fitted one time series with other time series. In this sense, we did not analyse uncertainty as one may do with GCM outputs generating ensemble runs of several thousands. If one think in terms of how good CHESS data represent climate variables, we checked with our colleagues and they were of the opinion that the main weakness in the CHESS data was the downscaling of MORECS data from 40km to 1km, which may cause some variables to be overestimated in some parts of the UK; however, we had no sites located in those parts. Given the models performed reasonably well at predicting the observed water temperatures (conditional R-squared within 0.84-0.96), we consider that any uncertainty is acceptable and does not affect the overall analysis massively. In addition, with multi-level modelling, confidence intervals, although they can be calculated, are not considered as meaningful as for standard regression models.

Change #11: None.

Comment #12 P.6,l.20-22: what is the reasoning behind investigating seasonal averages? Why only these? What is their ecological relevance? What about seasonal average daily (or 7-daily) maxima and minima? Would these not be more instructive? Just wondering.

Response #12: Ecological relevance is with regards to phenology (clarification added in the Introduction). In addition, research fitted within a wider research on seasonal hydro-climatic patterns (eg Laize & Hannah, 2010). Minima and maxima would be of interest if investigating topics like lethal thresholds).

Change #12: Clarification regarding phenology added to Introduction (P3, L30-31).

Comment #13 P.7,l.4-5: where is this section? I cannot find it. This is relevant information and needs to be shown.

Response #13: This information is actually in the Results section, but in response to Comment #25 below (and also to a similar comment from Reviewer #2), we followed the reviewer's suggestion that this should appear earlier in the manuscript, and moved it to the Data section.

Change #13: See Comment #25 below.

Comment #14: P.7,l.5: what is meant by "permeability"? permeability of what? How was it determined?

Response #14: We meant catchment permeability in the sense of flashy impermeable catchments vs groundwater-fed catchments. It is characterised by using the catchment base flow index (BFI; described later in the text).

Change #14: We clarified this point (P7, L15-16 in revised manuscript).

Comment #15: P.7,l.6: not clear what is meant by this sentence.

Response #15: These basin properties are generally recognised in UK studies (cited studies and many others) as modifiers of climate-hydrology associations.

Change #15: The sentence was reworded to improve clarity (P7, L16-17 in revised manuscript).

Comment #16: P.7,l.23ff: how was the spatial correlation structure between sites along the same rivers accounted for? What was the flow distance between the sites closest to each other?

Response #16: It was taken into account by using multi-level modelling. As explained in the method section, the multi-level models were specified with 3 levels: data, data at a site, sites on a river. With one level representing rivers, the multi-level models were able to take into account the fact that two sites on the same river may have more similar records than two sites on different rivers due to their physical linking.

Change #16: None.

Comment #17: P.7,l.24: it should at least be mentioned that linear models, in particular for the air-water temperature relationship, are oversimplifications and that for example logistic models can much better account for effects such as evaporative cooling (e.g. Mohseni et al., 1998).

Response #17: Agreed.

Change #17: We added this point to Section 3 Methods (P8, L17-20 in revised manuscript), including the reference to Mohseni et al. (1998).

Comment #18 P.7ff, sections 3.1, 3.2: I found this quite hard to follow. I would like to encourage the authors to invest some more effort to describe this critical part of their analysis more clearly.

Response #18: We reviewed these sections and clarified the more confusing points (the reviewer's comment is not specific in this regards).

Change #18: Section 3.1 and 3.2 substantially edited and expanded (see specific edits as Track Changes, P8-11 in revised manuscript).

Comment #19 P.8,l.21-25: so how were the various combinations tested? Stepwise regression or best sub-set regression or some other method? What is AICc? How was it corrected for small-size datasets?

Response #19: As described in the text, all combinations were fitted (programmatically) and their AICs calculated. No stepwise regression or sub-setting was done. The model with the lowest AICc was retained. We edited the text to clarify that process. AICc is one standard option in R; the difference with AIC is a slightly modified formula putting more penalty on the number of parameters than with normal AIC. This is the suitable thing to do in this study (there are accepted rules about number of predictors v sample size). This is actually a minor technical detail, which has been included for completeness. Detailing AICc any further would require to detail AIC, which is itself fully described in Akaike (1974); AIC and AICc are nowadays standard tools of the trade.

Change #19: Text edited to clarify selection process (P10, L7-15 in revised manuscript).

Comment #20: P.10, section 4.1: not clear which explanatory variables were used in the individual models. All?

Response #20: All predictors were used. Predictors with RI equals to 1 were used in all models. Other predictors (with RI < 1) were used in some of the models constituting a set of best models.

Change #20: None.

Comment #21: P.11,l.1: what does "adequately" mean? Please provide R2 and p-values.

Response #21: When using multi-level modelling (and moreover within the multi-model inference (MMI) framework), $R^2$ as commonly featured for regressions are actually not suitable, hence the current choice of showing observed vs predicted plots only. However, to address the reviewer's point, there is an alternative $R^2$ for multi-level models ("conditional $R^2$") by Nakagawa and Schielzeth (2013), which we calculated and added to Section 4.1 (reference to the paper was added too). Word "adequately" was removed. Regarding p values, their conceptual equivalent within a MMI framework is the predictor relative importance RI used in the paper. A sentence in the Methods Section 3.2 and one sentence in Section 4.2.1 have been added to highlight this, and clarify that higher RI means more significant predictors.

Change #21: A paragraph presenting conditional $R^2$ (including reference) and providing values for present studies was added to Section 4.1 (P12, L27-29 and P13, L1-2, in revised manuscript). Sentence (P13, L5-9 in revised manuscript) was edited (in particular, we removed "adequately"; see Track Changes). One sentence added to Section 3.2 (P11, L21-25 in revised manuscript). One sentence added to Section 4.2.1 (P13, L12-14 in revised manuscript).

Comment #22: P.11,l.3ff, section 4.2: one thing that is completely missing here but that may be of considerable relevance is the potential collinearity (or correlation) between the predictor variables, which can potentially result in highly unstable and misleading model results. It will therefore be necessary to quantify the collinearity and evaluate to which degree it actually influences the results.

Response #22: We were aware of possible collinearity issues and this was one of the reasons to use MMI. Collinearity gives inflated standard errors of parameter estimates. Approaches like MMI are fairly robust to even high levels of collinearity (see for example, Feckleton, 2011; Grueber et al, 2011); simply put, if there is high collinearity between two variables, then they do not appear in the same model, and do not force standard errors up. In addition, in this case, correlations between predictors were below 0.5 (Pearson) for most pairs, and the more highly correlated pairs were still within what is generally regarded as reasonable by statisticians. In the results/discussion, we aimed to take into account the implications of predictors co-varying when interpreting observed patterns.

Change #22: None.

Comment #23: P.12,l.2: please clarify: are the percentage contributions in fact the proportions of the explained variance?

Response #23: For each record in the dataset, WT was predicted using the average model coefficients from Table 3, then the % contribution of each predictor to predicted WT was calculated (ie we made a time series of WT predictions and predictor % contributions).

Change #23: We edited the text to clarify the above (P14, L11-15 in revised manuscript).

Comment #24: P.12,l.4, figure 5: please provide a unit for the y-scale in the figure. The unit of the x-scale (%) seems to be wrong here. In addition, please be more specific: % of what?

Response #24: The y-axis label ('%') was erroneously placed on the x-axis. Captions explain what the % are for both sets of plots (we found that trying to abbreviate the % definition to fit it as an axis label did not really improve the figures).

Change #24: Figure amended (% relocated to y-axis).

Comment #25: P.13,l.19ff: this needs to go into the methods section. Please also clarify why exactly these properties were chosen and provide a table with the relevant values.

Response #25: The section on basin properties was originally in Data and Methods but, because part of the analysis was used to confirm the selection of FEH properties (out of the available 19 properties), and because the Data and Methods sections were already quite long, we felt that it could be considered part of Results instead. We appreciate that this can actually be confusing, and that readers would more likely expect this information earlier in the manuscript. We therefore opted to move a significant part of Section 4.3 to Data Section 2.4 Basin properties. Regarding a table of values, we assume the reviewer means the actual property values for each site; we included in the text of Section 4.3, the ranges of values for each property across the 35 sites. We believe this could be a good compromise in terms of information vs manuscript length, and would favour keeping the manuscript as it stands.

Change #25: Text from Section 4.3 moved to Section 2.4 (P7-8 in revised manuscript); see Track Changes for details.

Comment #26: P.13,l.20: elevation not only related to wetness but clearly also to air temperature

Response #26: We added this comment when elevation is introduced.

Change #26: Comment added (P7, L31 in revised manuscript).

Comment #27: P.13,l.26: area is proxy for discharge and thus for thermal capacity, but is also linked to elevation

Response #27: We added this when the property is introduced. Note that this was already mentioned in the Discussion.

Change #27: Comment added (P8, L1-2).

Comment #28: P.13,l.27: what is the reasoning behind using HOST/permeability? What is it
expected to explain?

Response #28: We were expecting groundwater-fed catchments to behave differently from
impermeable ones (eg temperature regime influenced by groundwater inputs). We added this
when the property is introduced. It was already covered in the Discussion.

Change #28: Comment added (P8, L4-6).

Comment #29: P.14,l.8: please also provide the individual p-values!

Response #29: The models here were selected using MMI as per the main WT models. But,
unlike for the main models, for which only the average model was featured, Table 6 lists all
the models (and their R-squared and AIC) included in a MMI model set. This may give the
impression the models were fitted using traditional approaches (eg removing predictors with
high p values as not significant) although it was not the case. Please also see response #21 to
similar comment re using MMI and selection with an information criterion (ie p values are not
relevant).

Change #29: None.

Comment #30: P.14,l.14-15: this is a sweeping generalization which needs to be toned down

Response #30: Agreed.

Change #30: Sentence revised (from P16, L24-25).

Comment #31: P.14,l.16: why should there be more small basins at higher elevations?
Channel formation does not have anything to do with elevation, but rather with contributing
area and local slope. There may be some correlation with elevation but it is not generally
valid as posed here. what, however, is true is that, necessarily the opposite is true: there are
more larger basins at lower elevations.

Response #31: Agreed.

Change #31: Sentence revised (P16, L26-28).

Comment #32: P.16,l.5-6: this is possible, but not sufficiently substantiated by data here. I
would argue that it is equally likely the indirect correlation is merely a model artifact without
physical meaning (and potentially related to collinearity).

Response #32: An early version of the manuscript actually made that very point. We re-
instated it but kept the possible physical explanation as well.

Change #32: Point above inserted in text (P18, L19-20 in revised manuscript).

**Response to Reviewer #2**

Comment #1: This article explores basin and climatic drivers of stream temperatures across the UK. While the authors do a nice job throughout stating what is novel with respect to the study, I have a hard time finding some of their results novel. They show that air temperature, and solar radiation, drive heat fluxes throughout the year. Their findings fall in line with 30+ years of stream temperature research. A potentially novel result is the inclusion of different climatic factors, and the modeling style that they use to include these factors. However, it is not clear what this information adds to predictive capacity for stream temperature across the UK. Does including these variables mean there is greater explanatory power? Tertiarily, they also relate models to basin properties. However, the basin properties that were included are not well described in the paper, and end up feeling tangential to the other results. I'm left wondering where the model(s) perform(s) well, and where they performs poorly, and how performance changes across different scales. Can this approach we used to improve modeling of stream temperatures? This is mentioned briefly at the end. As it stands, showing that models identify climatic variables as important seems to confirm what we already know. Showing, again, that basin properties influence these results is also potentially not new. I'm also left wondering about some of the implications of their data (in terms of temporal and spatial extent) for their conclusions. Overall, this is clearly a well-developed idea that will advance stream temperature research, but I am left feeling confused about broader implications, the sites in question, and whether this type of approach gets us any closer to improving our empirical modeling of stream temperatures.

Response #1: It is worth clarifying that we did not aim to produce a better predictive model of water temperature, but rather, the modelling exercise was a mean to gain better understanding of the large-scale spatial and temporal variability in climate–WT associations, and of the influence of basin properties on these associations. The modelling techniques (multi-level modelling and multi-model inference) are definitively novel in their application to water temperature. In particular, we could analyse data both from at site scale and at national scale at once with multi-level modelling. The sites covered a reasonably wide range of catchment types. The combined wider spatial patterns and site-specific responses related to basin properties help unravelling the relative influence of climate vs land surface control across scales.

Change #1: None.

Comment #2: Results, especially in tables and figures, are not presented in a way that enables easy interpretation by the reader. Table 6 means nothing to anyone but the authors. Table 5 – why is the FEH descriptor included, except for reference to Table 6? Why were the selection of descriptors used? Greater insight on which descriptors were included would be helpful. Section 4.3 for instance, refers to the abbreviations of FEH variables, but it would be much fewer words to just state the actual variables in text, and indicate FEH variables in parentheses

Response #2: We think this comment stems from the fact that we introduced the basin properties in the Results rather than in Data and Methods. As we stated in response to Reviewer #1 Comment #25, we moved the bulk of Section 4.3 back to Data, thus streamlining and clarifying which basin descriptors we used. Re Table 5, we included the FEH descriptor to highlight the fact that the basin property (eg elevation) is characterised by a specific descriptor (eg ALTBAR). Then, since these descriptors are indeed used for the results featured in Table 6, we thought it would help readers to make the connection. We believe we explain Table 6 clearly enough in the text. Regarding the latter point of FEH descriptors abbreviations, they are actually not abbreviations as such but short names (except for BFIHOST), so their explicit names is already given (eg ALTBAR = mean basin elevation above sea level). To clarify things, we swapped order of full name and FEH short name.

Change #2: Text from Section 4.3 moved to Section 2.4 (P7-8 in revised manuscript); see Track Changes for details; descriptors full names and short names swapped (P7-8).

Comment #3: The introductions to each section are not helpful, but I leave this up to the authors. I find that they detract from the reading of the manuscript.

Response #3: Experience with past papers showed that some readers benefit from these section introductions. We are inclined to keep them unless there is a need to reduce the length of the manuscript.

Change #3: None.

Comment #4: Sites with very different time scales of measurement where included. I get why this was done – there is not a lot of stream temperature data (a problem I am also having!). However, I'd like to know more about what is the effect? Were sites with 15-minute versus weekly and monthly data treated differently? With so many sites, it would be worth testing if 15-minute data were treated in the same way as weekly or monthly sites, what the effect on conclusions would be? If sites from weekly/monthly data were excluded, are conclusions different?

Response #4: The discrepancy in data time scales is handled by using multi-level modelling. So, for example, if sites based on 15-min data behaved differently from sites based on weekly data, the model would correct for that. However, it would not explicitly investigate what the effect of one over the other. We added a mention of this as possible future research in the Conclusions.

Change #4: Sentence added in Conclusions (P22, L13-16).

Comment #5: Unclear what kind of variability in terms of basin/river properties your paper explores – a figure to this effect would be a good contribution. For instance, where else would your results be comparable to? This would be helpful to know both in terms of stream temperature regime and basin properties.

Response #5: In Section 4.3 (moved to Section 2.4 in revised manuscript), we have included a paragraph giving the ranges of basin properties for the 35 sites. They do provide a fairly wide range of basin types in the UK. The original data sources include a mix of lowland permeable (eg from LOCAR, Tadnoll; the UK has most of its aquifers in lowland regions) and impermeable basins, upland impermeable (eg form Plynlimon), as well as small to medium basin sizes. Sites from the AWMN cover all types. The gap in coverage may be in terms of large basins (ie >1000 km2) but these are far less common in the UK than in other countries. Similarly to our response to Reviewer #1 Comment #25, we think that the manuscript as it stands provides a good compromise between information vs length.

Change #5: None.

Comment #6: Magnitude of fluxes depend not only on climatic variables, but also on water
temperature. Is model able to include this interaction, as it is a key determinant of
evaporation/condensation and convection/conduction?

Response #6: As it stands, the models are at their core linear regression with water
temperature as the dependent variable. Feedbacks cannot be built-in. This would require a
different type of method, possibly bespoke models handling iterative calculations. However,
we explored this with the outputs from Fig 5 showing how the contributions of different
predictors change with water temperature.

Change #6; None.

Comment #7: Need more information on descriptors. They're included haphazardly. Don't
even know which predictors are included in the model.

Response #7: See responses to Reviewer #1 Comment #25, Reviewer #2 Comment #2 re
Section 4.3 moved to Data. We believe the changes to the way basin properties are presented
address this.

Change #7: Text from Section 4.3 moved to Section 2.4 (P7-8 in revised manuscript); see
Track Changes for details.

Comment #8: Pg 3, Lines 10 – 20 – variables should have subscripts

Response #8: Agreed.

Change #8: Text amended as suggested (P3).

Comment #9: Pg 3, Line 28 – misplaced comma

Response #9: Agreed.

Change #9: Comma moved to its proper position.

Comment #10: Pg 4, line 6 – consider the role of basin properties with respect to what?
There's several papers in the US that have investigated the role basin properties may play in
determining the stream temperature regime – they do so from an empirical perspective

Response #10: We clarified we consider the role of basin properties with regards to stream
temperature, and added the point about the empirical approach (also see response to see
Reviewer #1 comment #5).

Change #10: Sentence (P4, L8-9 in revised manuscript) amended.

Comment #11: Pg 5, line 3 – it's not clear to me what you mean by 'not losing any
information'

Response #11: This refers to the loss of information due to class-level averaging, a common
step with classification-based analyses (already covered in a more detailed way on Page 4).
We clarified this point.

Change #11: Clarification inserted (P5, L7 in revised manuscript).

Comment #12: Pg 6, section 2.2 – what impacts do you think using a 1km square meteorological dataset may have on your proposed conclusions? Are there any sites where microclimate could play a role?

Response #12: We agree there may be micro-climate effects (one co-author published several papers based on field site monitoring showing the impact of shading, etc.), but the focus of this study was the wider spatial patterns, which is quite novel, so we did not investigate this further. In addition, based on the information we had, we do not think there was any site where micro-climate effect was conclusively present. However, we mention this in the discussion (for example, highlighting how shaded river may behave differently from non-shaded). We added a mention of this point in the conclusion.

Change #12: Sentence added in Conclusions (P22, L29-33).

Comment #13; Section 3.2 was difficult to follow and written confusingly. Comments were included in parentheses and not explained fully. The importance of AIC weights was introduced, but there was little explanation of what this value tells the reader (does 'relative importance' mean a better model? More trustworthy model?)

Response #13: This particular section has been revised and streamlined to provide more clarity (see response to Reviewer #1 Comment #18).

Change #13: Section 3.1 and 3.2 substantially edited and expanded (see specific edits as Track Changes, P8-11 in revised manuscript).

Comment #14: Some missing words in section 3.3

Response #14: There was in fact an extra noun, which should have been deleted. Section has been checked and revised.

Change #14: Unnecessary word deleted (P11, L28).

Comment #15: Page 10 line 24: why was no predictor included for spring?

Response #15: As explained in that paragraph, for spring, the model getting the lowest AICc (ie the best model) was the model with random intercept only (ie the only difference between sites is with regards to the mean water temperature; all sites have the same response slope for all predictors). Therefore, no predictor was included as a random effect. We added a reminder that spring is a random intercept only ML model. We also expanded Section 3.1 to give more clarity regarding multi-level modelling, which in turns would help readers to understand what is meant in this paragraph.

Change #15: Reminder that spring is a "random intercept only" ML model (P12, L23). Section 3.1 expanded to give more clarity regarding random effects in multi-level models (P9, L22-28; see Track Changes for details).

Comment #16: Abbreviations make the results difficult to adjust – I know what short wave radiation is, but every time I see SWR, I get confused!

Response #16: We appreciate this problem, but we had to use abbreviations for the sake of conciseness (there are many references to the model predictors in the text), and following that, consistency required using these abbreviations all through out.

Change #16: None.

Comment #17: Pg 17 line 1: Most other studies only use AT because it so well predicts stream temperatures. While your models demonstrate association, how much better do they predict stream temperature than air temperature alone? Furthermore, you use gridded AT data, which is available everywhere. I find it much less likely that AT is unavailable at a site with a suite of other climatic variables.

Response #17: Our objective was not to build a better predictive model but to understand the various climate-stream temperature associations. As such, we do not claim that these particular models do necessarily better than AT-WT models (and did not investigate this), but that using other climate predictors in addition to AT could be informative in some cases. We added a few words to avoid misunderstanding on that point. Regarding the comment about AT being not available, it is true that if one uses gridded climate data, then AT is probably more likely to be available than some of the other variables: what we had in mind were field sites where air temperature measurements may be missing. However, this was a minor point however, so we deleted it to avoid distracting from the main message.

Change #17: Text inserted (P19, L12-13). Text deleted (P19, L18).

Comment #18: Pg 17, line 27 on – please rephrase out of list form

Response #18: Agreed.

Change #18: Text edited as requested (P20, L11-14).

Comment #19: Figure 4 should be improved – it is difficult to read axes. Model fits should be included.

Response #19: The size of the figure has been increased slightly to improve readability (figure has also been edited to amend one label, which was not displayed properly). Model fits (conditional R2) have been added to text. Given the conditional R2 are strictly speaking calculated for each model in a model set rather than for the average model (ie the average of all models in a model set), we think it is more accurate to leave the reader to gauge the average model fits visually than to add a mean R2 to the plots (see Reviewer #1 Comment #21).

Change #19: Figure amended. Paragraph giving conditional R2 inserted (P12, L27 to P13, L2, in revised manuscript).

Comment #20: Figure 5, please label the y-axes

Response #20: The y-axis label ('%') was erroneously placed on the x-axis.

Change #20: Figure amended.

[revised manuscript text omitted]

---

## Author Response (AR2)

Authors' response to reviewers.

Reviewer #1 Accept as is (no corrections)

Reviewer #2 minor revisions

Comment #1: The introduction is excellent but long (9 paragraphs). There is a fair amount of
theory/explanation that could potentially be moved to one of the sections describing the
methods. I would invite the authors to at least consider this.

Response #1: We reviewed the Introduction. We included quite a lot on the modelling method
because we neeed to to emphasize that this study used modelling techniques that are quite
novel for the field. We therefore decided to leave the text covering multi-level models as it is.
However, we moved the last sentence of the Introduction, which covers multi-model
inference (and includes the Grueber et al 2011 reference), to the Method section, thus
tightening the text and shortening it slightly.

Change #1: Last sentence of Introduction (P5 L10-13) deleted, and text added to Methods (P8
L19-20).

Comment #2: I would like to see a bit more discussion of the limitations of examining such
large-scale spatiotemporal temperature patterns. The authors explain well the advantages of
these scales (indeed this is a distinguishing characteristic of the paper), but what about the
disadvantages (i.e. the paper limitations)? What is not captured by considering higher
frequency patterns in time (e.g. heat waves and the role on ecosystems and other services)?
Also, what is missed by not considering smaller scale spatial variability (i.e. thermal diversity
or thermal refugia – see Kurylyk et al. 2015). Some of this 'large-scale' mindset influences
the writing in places. For example, it is stated in P2, L28-30 that climate controls are most
important, while hydrologic controls are of secondary importance. I would argue that order is
(or at least can be) switched when it comes to thermal diversity.

Kurylyk BL, MacQuarrie KTB, Linnansaari T, Cunjak RA, Curry RA. 2015 Preserving,
augmenting, and creating cold-water thermal refugia in rivers: concepts derived from research
on    the    Miramichi    River    (Canada).    Ecohydrology    8(6)

Response #2: The study is indeed focusing on larger spatio-temporal scales, which we make
very explicit starting with the paper title (which includes such key words as "basin" and
"seasonal"), and as the reviewer points out, with the introduction. However, we thought it would be useful to emphasize this even more clearly in the introduction and added a sentence stating that we did not focus on higher frequency temporal patterns or smaller spatial patterns.

Change #2: Sentence added (P2 L26-28).

Comment #3: The results and discussion sections have unnecessary mini introductions that should be deleted (e.g. P15, L27 – P16, L3)

Response #3: Agreed.

Change #3: Mini-introduction of sections 4 and 5 deleted.

Comment #4: The discussion around P18, L23 addresses the relationship between the climate variables (and the WT sensitivity to them) and the basin conditions. I might be more careful with some of the wording to be clear that the authors did not look at long term climate sensitivity but rather how the seasonal climate data influenced the water temperature. For example, it is stated: "Thus, water temperature in 24 impermeable basins appears to be more sensitive to climate than in permeable basins". The data in the paper do not support that this is true on a multi-decadal basis but rather just a seasonal basis. Climate implies long terms. Perhaps they should reword 'climate' as 'seasonal climate data' or something like this.

Response #4: We reviewed Discussion 5.2 and Conclusions sections, changing 'climate' for 'seasonal climate data' in four locations, which clarify that other uses of 'climate' nearby (eg within same paragraph or sub-section) has to be understood in that sense (ie not long-term multi-decadal climate).

Comment #5: Table 2 and related text: Is it possible that precipitation was more a statistical indicator of streamflow (and thus thermal inertia) rather than an indicator of surface advective input? This seems like it could be especially true in headwater streams

Response #5: In Table 2, aimed to relate the climate variables from CHESS to an actual physical process. So, with that perspective, precipitation would be an advective input indeed; we left Table 2 and associated text unchanged as a consequence. However, the reviewer makes the point that precipitation, as a predictor in the models, could be a proxy for the effect of increased streamflow and thermal inertia. We added this point in the Discussion.

Change #5: Text added (P17 L24-25).

Minor comments

P2, L20, the '(a)' is misplaced in this sentence causing the sentence to read awkwardly. **Done**

P4, L1, 'rarely' should be moved before 'coordinated' **Done**

P9, L7 typo (similar) **Done**

P9, L13, missing word(s) at end of sentence **Missing word ('sites') added at end of sentence**

P17, L1 should be 'only partly causal' **Done**

P18, L28 I don't think 'damper' is the right word (or a word) **We checked and it is actually**

**the right noun (something that dampens) but used as an adjective; we replaced it with**

**'dampening'.**

Figure 1 caption should be slightly reworded. It is a bit confusing **Caption edited**

[revised manuscript text omitted]